



# Evaluation of the COSMO model (v5.1) in polarimetric radar space – Impact of uncertainties in model microphysics, retrievals, and forward operator

Prabhakar Shrestha[1], Jana Mendrok[2], Velibor Pejcic[1], Silke Trömel[1,3], Ulrich Blahak[2], and Jacob T. Carlin[4,5]

[1]Institute of Geosciences, Meteorology Department, Bonn University, Germany
[2]Deutscher Wetterdienst, Offenbach, Germany
[3]Laboratory for Clouds and Precipitation Exploration, Geoverbund ABC/J, Bonn, Germany
[4]Cooperative Institute for Mesoscale Meteorological Studies, University of Oklahoma, USA
[5]NOAA/OAR National Severe Storms Laboratory, Norman, Oklahoma, USA

**Correspondence:** Prabhakar Shrestha (pshrestha@uni-bonn.de)

**Abstract.** Sensitivity experiments with a numerical weather prediction (NWP) model and polarimetric radar forward operator (FO) are conducted for a long-duration stratiform event over northwestern Germany, to evaluate uncertainties in the partitioning of the ice water content and assumptions of hydrometeor scattering properties in the NWP model and FO, respectively. Polarimetric observations from X-band radar and retrievals of hydrometeor classifications are used for comparison with the multiple experiments in radar and model space. Modifying two parameters ($D_{ice}$ and $T_{gr}$) responsible for the production of snow and graupel, respectively, was found to improve the synthetic polarimetric moments and simulated hydrometeor population, while keeping the difference in surface precipitation statistically insignificant at model resolvable grid scales. However, the model still exhibited a low bias in simulated polarimetric moments at lower levels above the melting layer (-3 to -13°C) where snow was found to dominate. This necessitates further research into the missing microphysical processes in these lower levels (e.g., fragmentation due to ice-ice collisions), and use of more reliable snow scattering models to draw valid conclusions.

## 1 Introduction

Polarimetric radar networks provide an unprecedented database to evaluate and improve cloud microphysical parameterizations in numerical weather prediction (NWP) models. With the increasing availability and use of such modern remote sensing observations (also satellites, radiometers etc.) for NWP model validation, evaluation, and data assimilation, there is an increasing demand for cloud microphysics parameterization schemes to realistically approximate cloud microphysical processes and hydrometeor properties such as size distributions, partitioning into different classes/types, bulk densities, fall speeds etc. This is key for consistent forward simulations of cloud-related quantities across different measurement platforms and different parts of the electromagnetic spectrum, because these radiative transfer calculations depend critically on the particle properties and their spatial distributions. Errors herein can lead to inconsistencies in simulated measurements for the same cloud, which may cause adverse effects, for example, in data assimilation. Even with a single device like a polarimetric radar, there can be such





inconsistencies because different polarimetric parameters are related to different moments of the particle size distributions of modeled hydrometeor species.

The aforementioned inconsistencies can be larger above the melting layer, where uncertainty exists, among other things, in the partitioning of the total ice water content (IWC) across hydrometeor species—cloud ice, snow aggregates, graupel

and hail—in cloud microphysics schemes (van Lier-Walqui et al., 2012; Morrison et al., 2020; Thompson et al., 2021). This partitioning is often tuned in NWP models to prioritize and ensure good quality of the simulated surface precipitation. However, the simulated cloud microphysical processes aloft might deviate from reality (e.g., Lang et al., 2007; Fridlind et al., 2017; Han et al., 2019)). Lang et al. (2007) showed that removing dry growth of graupel in the 3D Goddard cumulus ensemble (GCE; Tao and Simpson, 1993) model reduced excess graupel production in the anvil and stratiform portions of the convective storm.

However, this led to excess snow production, which could be compensated for, by further decreasing the collection efficiency of cloud water by snow. These changes led to more realistic hydrometeor profiles compared to aircraft estimates, with smaller cloud ice particles dominating the upper portions and snow aggregates dominating near the melting layer. Similarly, Fridlind et al. (2017) also reported that model simulations with NASA Unified Weather Research and Forecasting (NU-WRF; Peters-Lidard et al., 2015) model underpredicted total ice number concentrations and overpredicted total mass peak (due to snow

domination) at $5-8\,\mathrm{km}$ height compared to aircraft observations for the stratiform outflow region of a mid-latitude squall line. Han et al. (2019) also reported that for the stratiform region of a mid-latitude squall line, most microphysical schemes in the Weather Research and Forecasting (WRF; Skamarock et al., 2008) model generally overestimate IWC above $7\,\mathrm{km}$ compared to aircraft retrievals, and underestimate IWC below $5\,\mathrm{km}$, where it generally increases towards the melting level in aircraft data. While in-situ measurements using aircraft are very valuable, these data are generally limited in spatio-temporal

context. However, the availability of continuous regional coverage of polarimetric radar data provides high-resolution (e.g., X-band radar) insights into cloud microphysical processes, which can be used to evaluate and constrain cloud microphysical parameterization schemes and further improve NWP models.

In this study, we focus on the COSMO model (v5.1) at convection-permitting km-scale resolution in combination with the two-moment bulk cloud microphysical scheme of Seifert and Beheng (2006). This scheme is currently a candidate for

operational implementation into the regional NWP model of the German Weather Service (DWD).

We follow two pathways to exploit the information content of polarimetric radar measurements for model evaluation and improvement: (1) microphysical retrievals from radar, and (2) calculating simulated polarimetric radar fields from model data using a forward operator (Ryzhkov et al., 2020). For example, hydrometeor classification algorithms (HCA) exploit multi-dimensional polarimetric radar measurements to indicate the dominant hydrometeor type in each radar bin (e.g. Straka et al.,

2000; Besic et al., 2016). HCA enable us to evaluate the representation of hydrometeors in numerical models and aid in tuning the microphysical parameterizations to better match observations. Polarimetric radar forward operators, on the other hand, generate synthetic observations from the models, which enable a direct comparison in observation space including signatures of microphysical processes (e.g. Andrić et al., 2013; Putnam et al., 2017; Snyder et al., 2017). However, uncertainty also exists in the forward operators due to assumptions of hydrometeor scattering properties (e.g., liquid-ice phase partitioning, shape,

orientation, density) which are not available from the model. Therefore, the main goal of this study is to evaluate the synthetic





polarimetric observations from the model, and constrain/quantify the above uncertainties in the model and the forward operator by exploiting the information content of polarimetric radar measurements.

This study focuses on uncertainties regarding the 1) partitioning of ice water content among different hydrometeor types in the cloud microphysics scheme and 2) assumptions of hydrometeor scattering properties in polarimetric radar forward operators using a hindcast numerical experiment setup for a widespread wintertime stratiform precipitation event over northwestern Germany. We argue that these types of questions benefit from a simultaneous look at forward operators, retrieval techniques such as hydrometeor classification, and cloud microphysics by a team of members from both the numerical modeling and radar communities. Such horizontally uniform events facilitate the direct comparison of observations with numerical simulations and offer additional pathways to reduce the noisiness of radar observations, especially for phase measurements (see Sect. 2.1).

The paper is structured as follows. Sect. 2 describes the observations and the case under investigation, the NWP model, the forward operator (FO), and the radar retrievals used for this study. A first model evaluation with current default configurations of microphysics and FO is presented in Sect. 3. Sensitivity studies with model and FO are discussed in Sect. 4. Finally, overall discussion and conclusions are presented in Sect. 5 and Sect. 6, respectively.

## 2 Data and Methods

### 2.1 Radar Observations

Spatially and temporally high-resolution polarimetric weather radar measurements provide the undisputed core information for an in-depth evaluation of NWP models. In addition to improvements in quantitative precipitation estimation, polarimetric radars provide insights into precipitation microphysics and the 3D distribution of hydrometeors. This study exploits measurements of the polarimetric X-band Doppler radar (BoXPol) in the city of Bonn, Germany. It is installed at 50.73052°N, 7.0716638°E on a 30-m tall building next to the Institute for Geosciences, Department of Meteorology, at the University of Bonn at 99.9 m MSL (see also Diederich et al., 2015).

Trömel et al. (2014) and Ryzhkov et al. (2016) introduced so-called quasi-vertical profiles (QVPs) to present polarimetric radar observations in a time versus height format. To generate QVPs, the azimuthal median is calculated from standard conical scans measured at higher elevation angles and the range coordinate is transformed into height. Here we use the 18° elevation scan with 100 m range resolution. One key advantage is the inherent noise reduction from the averaging process which sometimes enables the detection and quantification of small but meaningful vertical gradients in the polarimetric radar variables. Furthermore, QVPs are well suited to study the temporal evolution of processes and to directly compare with other mostly vertically pointing sensors and, in particular, with model simulations. QVPs represent the average conditions within the cone spanned by the radar scan with decreasing resolution with height. If the precipitation is not uniform within the cone, errors from the averaging process are larger, but the advantage of a significant noise reduction clearly dominates for microphysical studies of widespread stratiform rain as presented in this paper.

The variables—horizontal reflectivity $Z_\mathrm{H}$, differential reflectivity $Z_\mathrm{DR}$, and cross-correlation coefficient $\rho_\mathrm{hv}$ are masked where $\rho_\mathrm{hv} < 0.7$ to exclude clutter and bins without significant weather signal from the attendant QVP calculations. Calibration





offsets for $Z_H$ and $Z_{DR}$ are estimated following Diederich et al. (2015) and Pejcic et al. (2021b). After differential phase $\Phi_{DP}$
is masked where $\rho_{hv} < 0.95$ and $Z_H < 0\,\mathrm{dBZ}$, the measurements are smoothed with a median filter using a $1.1$-km moving
window (i.e. including 11 range bins). The melting layer detection algorithm of Wolfensberger et al. (2016) is applied to each
ray of smoothed $\Phi_{DP}$ and the identified bins are removed. A linear interpolation of $\Phi_{DP}$ across the melting layer is performed
that excludes the component of backscattering differential phase $\delta$ and enables the estimation of an average specific differential
phase $K_{DP}$ in this region. Least-squares fitting on a moving $3.1$-km window is used to estimate $K_{DP}$ based on the smoothed
$\Phi_{DP}$ without melting layer contamination. Finally, the QVP of $K_{DP}$ is calculated using the same azimuthal median approach.

We study an event of widespread stratiform rain passing the Bonn area and monitored by BoXPol on 16 Nov 2014 between
00:00 and 10:00 UTC. Fig. 1 shows the QVPs during this time, illustrating only moderate reflectivities $Z_H$ around 20 to $25\,\mathrm{dBZ}$
near the surface and attendant $Z_{DR}$ values typical of rain. The melting layer is observed at around $1.5\,\mathrm{km}$ and significant $Z_H$
values are observed up to $6\,\mathrm{km}$. Enhanced $Z_{DR}$ values up to 0.4 dB indicate pristine crystals near the cloud top, while decreasing
$Z_{DR}$ together with increasing $Z_H$ toward lower levels indicates ongoing aggregation or riming processes. Enhanced $K_{DP}$ values
above the melting layer are especially observed in the first half of the observation period, pointing towards enhanced number
concentrations of ice particles and thus ice water content (IWC), which is in line with higher $Z_H$ below the melting layer
and thus higher rain rates in the first half of the observation period. $\rho_{hv}$ is mostly close to 1 except for in the melting layer
where values decrease to 0.92 to 0.95, in line with statistics of polarimetric variables performed in stratiform precipitation
observations at X band (Trömel et al., 2019). Pristine crystals together with decreasing signal-to-noise ratio also results in a
$\rho_{hv}$ reduction near the cloud top.

## 2.2 The COSMO Model (v5.1)

This study evaluates the partitioning of total ice water content in the Consortium of Small-scale Modelling (COSMO) model
(Steppeler et al., 2003; Baldauf et al., 2011) with a 2-moment bulk microphysics scheme (Seifert and Beheng, 2006) (henceforth
SB2M). The extended version of the SB2M is used which includes a separate hail class described in Blahak (2008); Noppel
et al. (2010); van Weverberg et al. (2014). An additional dynamic saturation adjustment was turned on to reduce the time step
sensitivity of model microphysics and precipitation (Barrett et al., 2019). More details about the dynamical core and other
physical schemes are available from Baldauf et al. (2011).

SB2M predicts the mass densities $\rho_q$ and number densities $\rho_n$ of cloud droplets, rain, cloud ice, snow, graupel and hail,
which are the zeroth and first moments of the particle size distributions (PSD) that is assumed to follow a modified Gamma
distribution (MGD)

$$f(x) = N_0 x^\mu \exp(-\lambda x^\nu) \tag{1}$$

with $x$ being the particle mass and parameters $\mu$ and $\nu$ determining the shape of the distribution. The specific hydrometeor mass
$q$ and specific number $n$ can be derived by $q = \rho_q/\rho$ and $n = \rho_n/\rho$ with $\rho$ being the total density (air, vapor and hydrometeors).





The size-mass and velocity-mass relations of different hydrometeors are parameterized by power laws

$$D = a_g x^{b_g} \qquad (2)$$

$$v_T = a_v x^{b_v} \qquad (3)$$

with (maximum) particle diameter $D$, terminal fall velocity $v_T$ and parameters $a_g$, $b_g$, $a_v$ and $b_v$.

The shape parameters $\mu$ and $\nu$ of the MGD remain constant for each hydrometeor class and $N_0$ and $\lambda$ can be diagnosed from

the two prognostic moments. However, if rain is below cloud base in the sedimentation-evaporation regime, it $\mu$ depends on the mean diameter (Seifert, 2008; van Weverberg et al., 2014) to better capture the strong effects of these two processes on the shape of the rain size distribution.

To mitigate unphysical effects on the mean spectral particle mass $\overline{x} = q/n$ coming from the separate advection and sedimentation of $q$ and $n$, it is very important to impose some minimum and maximum allowable mass limits for $\overline{x}$ ($x_{min}$ and $x_{max}$) at

relevant places during the model time stepping. This is done by clipping $n$ so that $\overline{x}$ stays within $[x_{min}, x_{max}]$. For reference, all fixed parameters which were used in this study are summarized in Table 1.

The cloud droplet nucleation parameterization is based on the lookup table of Segal and Khain (2006), which parameterizes the number of activated cloud droplets just above cloud base depending on the updraft speed $w$, ambient cloud nuclei (CN) concentration, mean and standard deviation of an assumed log-normal dry CN size distribution, and aerosol solubility, based

on 1D rising-parcel simulations with a very detailed bin microphysical scheme. For this study, continental aerosol with CN concentration $N_{CN} = 1700 \times 10^6 \, \mathrm{m}^{-3}$, log-normal standard deviation $\ln(\sigma_s) = 0.2$, mean radius of aerosol size distribution $R_2 = 0.03 \, \mu\mathrm{m}$ and solubility $\epsilon = 0.7$ is used. $w$ is chosen to be the prognostic grid scale updraft because the table values already include subgrid effects of a disturbed turbulent flow. Similarly, the ice nucleation parameterization is based on Kärcher and Lohmann (2002) and Kärcher et al. (2006). The large-scale concentration of aerosols of this parameterization for heterogeneous

ice nucleation are chosen as $N_{dust} = 162 \times 10^3 \, \mathrm{m}^{-3}$, $N_{soot} = 15 \times 10^7 \, \mathrm{m}^{-3}$, and $N_{organics} = 177 \times 10^7 \, \mathrm{m}^{-3}$.

For ice-phase processes, interactions between different hydrometeors involving collisions (e.g., riming, aggregation, ice-multiplication) play an important role in the partitioning of the IWC. These interactions are parameterized using collision integrals and collision/sticking efficiencies (Seifert and Beheng, 2006), which are activated according to certain particle mean size and temperature thresholds. In this study, we focus on two of these parameters, motivated by the case study below.

The first parameter is the critical mean diameter of cloud ice particles $D_{ice}$ for conversion to snow through aggregation of cloud ice. If the mean cloud ice size

$$\overline{D}_i = a_{gi}(q_i/n_i)^{b_{gi}} \qquad (4)$$

is larger than $D_{ice}$, self collection leads to the production of snow, otherwise ice remains as ice. $a_{gi}$ and $b_{gi}$ are the size-mass parameters for cloud ice and $q_i$ and $n_i$ are its specific mass and number, respectively. Perturbations in $D_{ice}$ affect the ice to

snow partitioning and also the size of the resulting snow particles.

The second parameter is the temperature threshold $T_{gr}$ below which the production of graupel by riming of cloud ice and snow with supercooled rain is allowed. If $T < T_{gr}$, a certain part of the rimed cloud ice and snow particles are converted to the





graupel class, otherwise no cross-class transfer happens. This pathway for graupel production may also be slightly controlled by $D_{ice}$, because larger snow exhibits more riming.

As with most bulk cloud microphysical schemes without a prognostic melted fraction, SB2M most likely systematically underestimates the distances which melting particles may fall until melting completely (i.e., the melting layer thickness). This is because SB2M instantaneously transfers the amount of meltwater formed during one model timestep from cloud ice, snow, graupel, and hail to the rain class. As a consequence, the melting hydrometeors shrink too fast, fall too slowly, and completely melt too quickly in the SB2M. We acknowledge this limitation in the SB2M scheme.

## 2.3 Forward Operator

Evaluating output of NWP models with observations requires their data in a consistent and comparable parameter space, typically either in model space (e.g., hydrometeor mass and number densities) or observation space (e.g., radar reflectivity and further polarimetric radar variables). Conversion of the numerical model output into radar observables is done by means of a polarimetric radar forward operator. Particularly for model evaluation, it is important that the model and forward operator (FO) are consistent regarding parameters that affect the forward modeled observables. Many of these, including the phase partitioning of hydrometeors during melting, the shape and orientation of particles, and the heterogeneous microstructure of frozen hydrometeors, are insufficiently constrained by the direct model output, and assumptions need to be made. When implicit assumptions exist in the numerical model, it is advantageous to ensure that the FO makes use of equivalent assumptions.

In this study, we apply the Bonn Polarimetric Radar forward Operator (B-PRO v2.0) (Xie et al., 2016, 2021). B-PRO is a research-oriented polarimetric radar FO, which has been built onto an early, non-polarimetric version of EMVORADO (Zeng et al., 2016), the German Weather service's operational radar FO. Below we explicate the B-PRO implementation and settings directly related to polarimetry. Further aspects are detailed in Appendix A.

B-PRO calculates and outputs polarimetric radar parameters on the spatial grid given by the numerical model field input. That is, no beam integration and antenna pattern are taken into account, and hence the output $Z_H$ and $Z_{DR}$ are to be considered as unattenuated (or perfectly attenuation-corrected) variables. In addition, no simulated measurement errors are included, making the output variables take on their intrinsic values. A software interface between the COSMO model and B-PRO ensures consistent microphysics, specifically the same hydrometeor class–dependent particle size distributions and size-mass relations used in the SB2M in Eq. (1) and 2 and Table 1.

The shape and orientation of the hydrometeors, which are the primary properties that characterize anisotropic scattering, are not at all constrained by the COSMO model. Instead, different parameterizations are applied by the FO. Apart from the cloud liquid class, hydrometeors are modeled as homogeneous oblate spheroids. Their shape is described by the aspect ratio (AR), i.e. the ratio of the spheroids' semi-minor and semi-major axes. The spheroids are assumed to have no preferential orientation, with their maximum cross section parallel to the horizon on average, and with canting angles $\alpha$ out of the horizontal following a Gaussian distribution with a specified width $\sigma_{canting}$, i.e. $\mu_\alpha = 0°$ and $\sigma_\alpha = \sigma_{canting}$ (see Ryzhkov et al., 2011). The applied hydrometeor class–dependent parameterizations for the frozen hydrometeors are given in Table 2. For rain, the AR parameterization of Brandes et al. (2002) and $\sigma_{canting} = 10°$ is used following Ryzhkov et al. (2011). The shapes and





orientations of melting particles are derived as melting fraction–dependent weighted mean values of the respective frozen hydrometeor and rain (see Appendix A for details on the melting model). Scattering properties of the spheroids are calculated applying the T-matrix method for particles with a fixed orientation (Mishchenko, 2000) for canting angle $\alpha = 0°$, with the

properties of particles with an orientation distribution then derived using the angular moments method of Ryzhkov et al. (2011, 2013a).

Compared to Xie et al. (2016), several modifications to B-PRO have been implemented within this study. Melting particle shape and orientation calculations have been adapted to consistently follow the approach of Ryzhkov et al. (2011). For cloud ice, the fairly spherical and unoriented crystals (AR $= 0.9 - 0.7$, $\sigma_{\mathrm{canting}} = 40°$) resulting in insignificant polarimetric signatures

have been replaced by more oriented ($\sigma_{\mathrm{canting}} = 12°$) and more non-spherical particles following a shape parameterization by Andrić et al. (2013). Also, the range of sizes considered for cloud ice, representing single crystal particles in COSMO, has been largely extended (from the original upper integration limit, $D_{\mathrm{int}}^{\mathrm{upper}}$, of 200 μm to 4 mm) consistent with COSMO (mean) size limits of cloud ice and providing a better coverage of particle sizes contributing to the cloud ice bulk properties. The considered size range of snow, representing aggregated ice particles in COSMO, was also found to not sufficiently cover

the sizes contributing significantly to bulk scattering (both regarding single particle scattering properties as well as the PSD-predicted number density). Therefore, the size ranges of snow, graupel, and hail were extended. The calculation of effective density and volume-equivalent diameter of the spheroids, which are inputs to the particle effective refractive index and T-matrix calculations, respectively, from $D$, $x$, and AR have been revised and corrected for both frozen and melting hydrometeors.

A summary of the shape and orientation parameterizations used as a baseline in this study (B-PRO$_{\mathrm{def}}$) is given in Table 2.

The table further details implementations or choices for further FO parameters like the melting scheme, effective medium approximation (EMA; see also Appendix A), and particle size ranges.

## 2.4 Hydrometeor Classification Algorithm (HCA)

In this study we use the HCA of Pejcic et al. (2021a), hereafter referred to as HCA-Pejcic. In this two-step method, agglomerative hierarchical clustering (Ward, 1963; Grazioli et al., 2015; Ribaud et al., 2019) is first applied to the polarimetric radar

observations and $\Delta_z$ (i.e., the difference between observation height and 0°C level). With the sigmoid transformation used in Besic et al. (2016), $\Delta_z$ separates liquid and solid regions with a smooth transition around the freezing level height. The resulting clusters are identified based on state of the art HCAs (Dolan and Rutledge, 2009; Dolan et al., 2013; Zrnic et al., 2001; Straka et al., 2000; Evaristo et al., 2013) and merged into categories comparable to the model hydrometeor classes for rain, snow, cloud ice, graupel, hail, and wet snow. In the second step, a modified method of Besic et al. (2018) is used to derive

hydrometeor percentage (HP). Here, we use not only the centroids (mean values of the polarimetric moments) of the clusters as done originally, but we also calculate the covariance of the five dimensional observations. Instead of the exponential distribution used by Besic et al. (2018), we use a multivariate normal distribution for the determination of the HP. This allows us to use the calculated centroids and covariances to determine the shape of the individual hydrometeor probability functions in five dimensions without parameterization. Furthermore, the membership function–based HCA of Zrnic et al. (2001) and adapted

by Evaristo et al. (2013) (HCA-Zrnic) and Dolan et al. (2013) (HCA-Dolan) are also used for comparison. HCA-Zrnic uses





hydrometeor categories of vertically aligned crystals, horizontally aligned crystals, wet snow, dry snow, graupel/hail, rain/hail, hail, large drops, heavy rain, moderate rain and light rain. HCA-Dolan uses categories of big drops/melting hail, hail, high density graupel, low density graupel, vertically aligned ice, wet snow, aggregates, ice crystals, rain, and drizzle. In these two methods, theoretically calculated membership functions are determined for each hydrometeor type and for each polarimetric

variable and a temperature variable. As explained in Zrnic et al. (2001) and Dolan and Rutledge (2009), the dominant classes are then determined by the highest score calculated over the membership functions.

## 3 Problem description

The study is set up over the Bonn radar domain (Shrestha, 2021). With BoXPol at the center of the domain, it encompasses the northwestern part of Germany bordering the Netherlands, Luxembourg, Belgium and France. The topography is dominated by

the Rhine Massif, the Rhine valley and the northwest lowlands (Fig. 2). The model domain covers an area of approx. $340 \, \text{km}^2$ with a km-scale horizontal grid resolution. Eighty levels are used in the vertically stretched layers with a near-surface-layer depth of 20 m. The COSMO-DE analysis data from DWD at $2.8 \, \text{km}$ horizontal resolution is used to process the initial and lateral boundary conditions at hourly intervals. The diurnal scale simulation is initialized at 15 Nov 2014 00:00 UTC and integrated for 35 hours with a time step of 6 s. The model output generated at 5 min interval from 16 Nov 2014 00:00 UTC to

10:00 UTC is used for the analysis. The control run (CTRL) uses the default model parameters with the prescribed cloud/ice nucleation parameters (discussed in Sect. 2.2).

For consistent comparison to the QVPs from the radar observations, the outputs of the model and FO are also postprocessed to obtain synthetic QVPs using conical scans with $18°$ elevation angle along the vertically stretched model grid. This is achieved by generating a one grid cell–wide circular mask for each model level, whose diameter increases with height as a function of

elevation angle, and using this mask (each containing a minimum of eight grid cells) to estimate the median value of the model or FO data at that level.

Fig. 3a-c shows the QVPs of modeled ice hydrometeors from the CTRL run of COSMO. The hydrometeor population here is dominated by snow, which is primarily produced by self-collection of cloud ice that grows rapidly via aggregation. As the hydrometeors fall downwards, the mixing ratio of cloud ice ($q_{\text{i}}$) decrease gradually, while the snow mixing ratio ($q_{\text{s}}$) increases

rapidly until the melting layer. The melting layer here is defined as the region around the 0°C isotherm located around $1.5 \, \text{km}$. At this height, the cloud ice and snow aggregates in the presence of cloud water and rain also produce considerable amounts of graupel via riming. The graupel mixing ratio ($q_{\text{g}}$) peaks at this height and then gradually decreases as it melts producing rain, while falling downwards to the surface. The QVPs of modeled rain for the CTRL run are shown in Fig. 4a . The rain mixing ratio ($q_{\text{r}}$) increases gradually below the melting layer towards the surface as the meltwater fraction from graupel is transferred

directly to rain.





## 3.1 Evaluation of synthetic radar observations

Synthetic radar observations of the CTRL run are derived by running the FO with the B-PRO$_{\text{def}}$ setup using the model outputs. To minimize the FO computational cost, the domain was cropped to only cover the QVP extent, which is simply governed by the maximum diameter of the conical mask near the top of the precipitating system. FO calculations were performed for each 5 min step during $00{:}00-10{:}00$ UTC, and QVPs were produced from the FO radar variable fields using a conical mask (see above). The resulting QVPs are shown in Fig. 5.

The cloud ice-dominated upper levels show reflectivities up to $10\,\text{dBZ}$, similar to the observations (see Fig. 1a). The increase of $Z_{\text{H}}$ with decreasing height through the snow-dominated layers and towards the melting layer is somewhat stronger compared to the observations, with $Z_{\text{H}}$ reaching up to $30\,\text{dBZ}$ just above the melting layer compared to $\leq 25\,\text{dBZ}$ in the observations. The maximum $Z_{\text{H}}$ in the melting layer agrees fairly well with the observations, but the melting layer appears wider in the model-simulated radar data compared to the observations. This pattern continues below the melting layer, where the synthetic $Z_{\text{H}}$ is significantly higher and high $Z_{\text{H}}$ appear over much longer times.

Cloud ice, located at heights $>5\,\text{km}$, shows a clear polarimetric signature with $Z_{\text{DR}}$ from 0.2 up to $2\,\text{dB}$, $K_{\text{DP}}$ up to $0.2°/\text{km}$, and $\rho_{\text{hv}}$ slightly decreased below 1. This is qualitatively in agreement with the observations; quantitative comparisons are not meaningful at these heights due to significant uncertainties in the observations. The snow-dominated layers are characterized by an evident lack of polarimetric signals in the synthetic observations, in clear disagreement with the observations showing $K_{\text{DP}}$ values of $0.1-0.2°/\text{km}$ and $Z_{\text{DR}}$ values ranging from $0.2-0.5\,\text{dB}$. Within and below the melting layer, synthetic $Z_{\text{DR}}$ reaches a clearly higher maxima (2 to $<3\,\text{dB}$) compared to observations, although the range and frequency of $K_{\text{DP}}$ values agree fairly well there. Synthetic $\rho_{\text{hv}}$ are by far not similarly reduced in the melting layer as in the observations, but show similar qualitative patterns. Below the melting layer, $\rho_{\text{hv}}$ exhibits clearly lower values than in the observations.

The matter of synthetic $\rho_{\text{hv}}$ in most regions being very close to 1 while observations show reduced values can likely be explained by shortcomings in the FO assumptions on hydrometeor shape and orientation. $\rho_{\text{hv}}$ describes the correlation between the horizontally and vertically polarized returned radar signals and is sensitive to particle shape, composition, and orientation; hence, it provides information about the diversity of the scattering particles within the observed radar volumes (Kumjian, 2018). That is, overestimating $\rho_{\text{hv}}$ indicates too little variability of assumed hydrometeor shapes and orientations in the FO. Assuming hydrometeors of all categories and sizes to be homogeneous spheroids, with identical shapes at least for all particles of one category and size, as is the state of the art for the majority of polarimetric radar FOs, is clearly a simplification of the shape and microstructure variability of real-world hydrometeors. Simplifications are typical and necessary in modeling, however, the persistent overestimation of $\rho_{\text{hv}}$ suggests that the current assumptions oversimplify and produce too little variety in the structure of particles, and therefore fail to sufficiently reproduce observed $\rho_{\text{hv}}$ levels. Similar issues have been identified, e.g., by Ryzhkov et al. (2013b). This current lack of forward modeling ability regarding $\rho_{\text{hv}}$ limits the applicability of synthetic $\rho_{\text{hv}}$ in data assimilation and in observation-equivalent based model evaluation.



The lack of polarimetric signatures at the snow-dominated heights may be due to issues with the partitioning of cloud ice and snow in these layers, but also due to the forward operator struggling to correctly model the scattering properties of snow aggregates. This is analyzed in depth in Sect. 4.1 and Sect. 4.2, respectively.

Large regions of high $Z_H$ together with extremely high $Z_{DR}$ and comparably low $\rho_{hv}$ (breaking the general pattern of synthetic values overestimating the observed one) below the melting layer suggest issues with the underlying COSMO modeled hydrometeor fields. Comparison to Fig. 4a indicates that the "curtains" of high $Z_H$ and $Z_{DR}$ do not coincide with high rain mixing ratios, and hence are likely not due to rain. Instead, as suggested by the frozen hydrometeor mixing ratios (Fig. 3a-c) and their mean sizes (not shown), these are caused by graupel. This has also been confirmed by no-graupel runs of the FO (not shown) and has been observed during testing to occur independently of the choice of melting scheme settings in B-PRO. That is, the FO results strongly suggest an overestimation of the graupel occurrence below the ML.

## 3.2 Comparison of model-predicted and retrieved hydrometeors

Fig. 6 shows the retrieved dominant hydrometeor types with HCA-Pejcic, HCA-Dolan and HCA-Zrnic. The HCA retrievals are only available up to an altitude of about 4.5 km (Fig. 6), because $K_{DP}$ values became too uncertain at altitudes above (Fig. 1). When comparing the dominant hydrometeor types, it can be seen that mainly snow is identified at temperatures above zero degrees. HCA-Zrnic and HCA-Pejcic show predominantly cloud ice at the upper edges of the precipitation and only between 06:00 and 07:00 UTC; from 09:00 UTC onward, cloud ice is also classified down to the height of the melting layer. HCA-Dolan classifies no cloud ice except for very small isolated areas between 05:00 UTC and 07:00 UTC at approx. 2.5 km. In contrast to snow, only very small amounts of graupel are classified. Thereby, HCA-Pejcic shows between 01:00 and 04:00 UTC and 07:00 and 09:00 UTC, graupel occurred directly above the melting layer, which can be only partially confirmed with the sagging of the melting layer (very visible in Fig. 1 at $\rho_{hv}$ and $Z_{DR}$) as an indicator of aggregation and riming. At the same time steps, smaller portions of graupel are classified by HCA-Zrnic. HCA-Dolan classifies rain above the 0°C isotherm in these areas and HCA-Zrnic in the middle of the melting layer. Considering the scores calculated via the membership function in HCA-Zrnic and HCA-Dolan (not shown here), there are only minor differences between the scores for graupel, rain and wet snow in the same areas. Fig. 7 shows the hydrometeor percentages derived from the HCA-Pejcic. Below the melting layer, all HCA classify rain and the mixing ratio for rain is constant at 100% and changes with height to wet snow. The mixing ratio of graupel reaches up to 3 km altitude at 01:00 UTC, but with only small proportions below 40%. The proportions of cloud ice reach down to 3 km altitude between 01:00 and 03:00 UTC. In general, the different HCA indicate the dominance of cloud ice above 4 km, snow aggregates from 1.5 km to 4 km (with sporadic appearances of cloud ice), wet snow in the melting layer, and rain drops below the melting layer. Comparison between the above radar retrievals and the CTRL simulations show two main differences in the partitioning of the IWC above the melting layer: 1) Excessive graupel production above the melting layer which extends from 1 to 2 km, and 2) a low concentration of cloud ice above 4 km and an absence of sporadic increases in cloud ice concentration above the melting layer. While the above deficiencies in the modeled hydrometeors in the CTRL run can be also observed in the synthetic radar variables compared to radar observations, with a thick bright band in the melting layer, stronger reflectivity above the melting layer, and a lack of polarimetric signatures in the snow-dominated region, the





contribution of possible errors in the assumptions used in the FO can also not be neglected. So, additional sensitivity studies with the model and FO are conducted to better understand the uncertainties in the model and the FO, which are discussed in detail in the following sections.

## 4 Sensitivity studies

### 4.1 Model sensitivity

#### 4.1.1 Setup

Table 3 summarizes the list of sensitivity experiments conducted to account for uncertainties in the partitioning of the IWC. The experiments include using different combinations of $D_{ice}$ and $T_{gr}$. In the SB2M scheme, $D_{ice}$ controls the aggregation of cloud ice, thereby affecting the production of snow and depletion of cloud ice. Aggregation of cloud ice is the primary source of snow production above $6 \, \text{km}$, which then further grows in size by aggregation of snow or between snow and cloud ice below. Similarly, $T_{gr}$ controls the riming of cloud ice and snow with supercooled rain drops, and hence affects the production of graupel and depletion of snow above the melting layer.

#### 4.1.2 Results

First, the model precipitation from the sensitivity runs was compared to the rain gauges available over Bonn, Germany. In general, the 10-hour accumulated model precipitation for all runs is similar compared to the gauge measurements (see Fig. 8). However, it is important to note that any perturbations in the model parameters can influence the spatial pattern of the precipitation. This can strongly influence the grid-scale comparison of precipitation between model and point observations. So, the domain average precipitation is also shown in Fig. 8a. A t-test was also conducted to check whether the difference in domain average precipitation between the CTRL and sensitivity runs were statistically significant (see Table 4). At native grid resolution, the difference is statistically significant ($p < 0.05$), while at actual model resolvable scales (e.g, $10 \, \Delta x$ used here), the difference is statistically insignificant ($p > 0.05$). However, both similarities and differences in the microphysical processes for rain production in the sensitivity runs can be observed. For example, $q_r$ for EXP1 is qualitatively similar to the CTRL run, but EXP2 and EXP3 differ (Fig. 4). For EXP2 and EXP3, a steady sharp increase in $q_r$ near the melting layer can be observed, while the $q_r$ gradient changes with time and appears more disorganized in CTRL and EXP1 runs.

Fig. 3d-f shows the QVPs of modeled frozen hydrometeors for model sensitivity experiment EXP3. The increase in $D_{ice}$ for cloud ice self-collection, as used in EXP1 and EXP3, substantially alters the hydrometeor population above $4 \, \text{km}$. Cloud ice now dominates above this height, while snow aggregates dominate below until the melting layer. The change in the partitioning of cloud ice and snow aggregates in the mid-levels ($>4 \, \text{km}$), however, has no significant effect on the graupel mixing ratio below. The decrease in $T_{gr}$ for graupel production, as applied in EXP2 and EXP3, only prohibits most of the graupel formation near the melting layer. Subsequently, the snow hydrometeors stretch further downward relative to the CTRL run and below the melting layer, eventually melting and producing rain. The absence of graupel production also has no effect on the hydrometeor





partitioning in the upper layers. However, the change in the source of ice hydrometeors to form rain drops via melting directly modulates the $q_r$ profiles below the melting layer (see Fig. 4). This could be attributed to the differences in the sedimentation
velocity and time-scales of melting for graupel and snow.

The change in the partitioning of frozen hydrometeors also leads to changes in the partitioning of cloud water and rain near the melting layer. Fig. 9 shows half-hourly averaged QVPs of hydrometeor mixing ratios at 0400 UTC for the CTRL and the sensitivity experiments. For EXP2 and EXP3, there is a general decrease in cloud water mixing ratio ($q_c$) with reference to the CTRL run. In the absence of graupel production, the sharp increase in $q_r$ near the melting layer due to melting of snow can also
be clearly observed for EXP2 and EXP3. In the time-averaged QVPs, the change in the partitioning of the cloud ice and snow aggregates in the mid-levels is also clearly visible between the CTRL and the sensitivity experiments. For the QVPs in Fig. 9, the mean size of graupel around the vicinity of the melting layer is around $1.5 - 2\,\mathrm{mm}$ for CTRL and EXP1. The mean size of rain is qualitatively similar in all runs ($<1\,\mathrm{mm}$), except that it increases rapidly near the melting layer for EXP2 and EXP3. Also, in these experiments snow aggregates stretch further downward below the melting layer as discussed above, with further
increase in mean size (from 2 to $3.5\,\mathrm{mm}$).

### 4.1.3   Effects in FO output

QVPs of synthetic radar observations for model experiment run EXP3 using the B-PRO$_{\mathrm{def}}$ setup are shown in Fig. 10. Since the effects of $D_{\mathrm{ice}}$ and $T_{\mathrm{gr}}$ changes are mostly independent and occur at different heights, their individual effects on the FO modeled observations can be, and are in the following, discussed for the FO output from the combined model experiment EXP3
only.

The change of the dominant hydrometeor from snow to cloud ice in the mid-levels resulting from the increase in $D_{\mathrm{ice}}$ between the CTRL and EXP1/3 runs leads to very little changes in $Z_{\mathrm{H}}$. However, the change in partitioning of cloud ice and snow above $4\,\mathrm{km}$ (increased amounts of cloud ice and reduction of snow) cause significantly intensified polarimetric signals with large regions of $Z_{\mathrm{DR}} > 2\,\mathrm{dB}$ and $K_{\mathrm{DP}} > 0.4°/\mathrm{km}$.

The reduction (or removal) of graupel resulting from the changes in $T_{\mathrm{gr}}$ largely remediates the "curtains" of high $Z_{\mathrm{H}}$, extreme $Z_{\mathrm{DR}}$, and depressed $\rho_{\mathrm{hv}}$ below the melting layer. It leads to a more well-defined, distinct melting layer signature and only leaves streaks of enhanced polarimetric signals, now exclusively resulting from rain, both much better in line with the observations. However, it also eliminates the melting layer signal in $K_{\mathrm{DP}}$. The latter might be a result of too little graupel now existing around the $0°$C level. It can, though, also be due to the already discussed lack of polarimetric signals from snow.

## 4.2   FO sensitivity

### 4.2.1   Setup

Weakly or unconstrained assumptions in the FO introduce uncertainties in the forward modeled observation parameters, also known as forward model error and are considered as one type of representation error in data assimilation (Janjić et al., 2018). Forward operator errors translate into errors and biases of the simulated radar variables, challenging the use of FOs in both





model evaluation and data assimilation. In order to study the forward operator uncertainty, specifically with respect to the polarimetric radar parameters, we set up a variety of B-PRO runs with perturbed assumptions in the polarimetry-relevant microphysics, i.e. the shape and orientation parameterizations.

The shape and orientation of rain drops are considered to be comparably well known, hence rain is not considered in this polarimetry sensitivity study. Also, the studied case does not contain any significant amounts of hail (neither in CTRL nor the

different experiment model run setups), therefore analysis of hail sensitivity is also skipped. For the remaining ice hydrometeor classes both shape and orientation have been varied, described by the parameterizations of the oblate spheroid aspect ratios and the canting angle distribution width, respectively. For all three classes of ice, the shapes and orientation reported in the literature, derived from a range of methods, including in-situ imaging of particles (e.g. Garrett et al., 2012), and over different atmospheric situations, vary strongly. From the literature, we have compiled a set of AR and $\sigma_{\text{canting}}$ values that cover these

reported ranges. For AR, we have selected three parameterizations providing a high, medium, and low AR (i.e. high(er) to low(er) sphericity) case for each hydrometeor class. While AR is often parameterized as a function of particle size, $\sigma_{\text{canting}}$ is mostly estimated as a constant value (one exception is Wolfensberger and Berne (2018), who described the $\sigma_{\text{canting}}$ of snow and graupel slightly decreasing with size). Here, we use a set of two constant $\sigma_{\text{canting}}$ per hydrometeor class, a high and a low value, corresponding to weaker or stronger degrees of orientation. Polarimetric FO calculations have been performed for each

combination of AR and $\sigma_{\text{canting}}$. A summary of all setups is given in Table 5.

In contrast to the synthetic QVP for the CTRL and EXP3 cases shown above, where B-PRO was run over the QVP extent, the sensitivity B-PRO calculations have only been performed for a single model column at the grid point location of the BoXPol radar, due to the high computational costs of the polarimetric FO. Since the precipitation system for this case study is horizontally homogeneous, the resulting single-column profiles are in general in good agreement with the full-domain QVPs

(compare Fig. 11), and conclusions drawn from comparing single-column FO results with QVPs can be considered robust.

### 4.2.2   Results

Results of the B-PRO sensitivity calculations are shown in Fig. 11 and Fig. 12 based on the CTRL and EXP3 model experiment output, respectively. For an easier comparison of the sensitivity, results are presented as median profiles over the time $00:00 - 10:00$ UTC. In addition to the B-PRO sensitivity run profiles, corresponding median QVPs from the full-domain run using the

B-PRO$_{\text{def}}$ setup (in grey) and from the BoXPol observations (in black) are included. Full-domain median QVPs closely follow the single-column profiles of the B-PRO$_{\text{def}}$ setup (bold lines) demonstrating the suitability of the single-column approximation.

Reflectivities $Z_{\text{H}}$ are found to be insensitive to the changes in all of the hydrometeor classes shape and orientation assumptions. As expected from scattering theory, polarimetric signals increase with increasing non-sphericity (i.e. lower AR) and higher degree of orientation (i.e. lower $\sigma_{\text{canting}}$). Shape and orientation effects on the radar variables used here are difficult

to disentangle since both decreasing AR (increasing non-sphericity) and decreasing $\sigma_{\text{canting}}$ (i.e. higher degree of orientation) generally lead to an increase in $Z_{\text{DR}}$ and $K_{\text{DP}}$. This renders independent evaluations or retrievals of shape and orientation challenging. Besides, the combined effects of AR and $\sigma_{\text{canting}}$ are not necessarily linear, but might instead amplify each other. This is, for example, observed in the case of snow (Fig. 11, middle row and Fig. 12, bottom row), where the change in $Z_{\text{DR}}$




and $K_{\mathrm{DP}}$ between the $\mathrm{AR_{mid}} + \sigma_{\mathrm{high}}$ case and the $\mathrm{AR_{low}} + \sigma_{\mathrm{low}}$ case (orange solid to green dashed) are clearly higher than the
changes from $\mathrm{AR_{mid}} + \sigma_{\mathrm{high}}$ to $\mathrm{AR_{low}} + \sigma_{\mathrm{high}}$ (orange solid to green solid) and $\mathrm{AR_{mid}} + \sigma_{\mathrm{high}}$ to the $\mathrm{AR_{mid}} + \sigma_{\mathrm{low}}$ (orange solid
to orange dashed) combined. The $Z_{\mathrm{DR}}$ and $K_{\mathrm{DP}}$ for the snow $\mathrm{AR_{low}} + \sigma_{\mathrm{low}}$ case are significantly higher than for all other snow
sensitivity setups, which cluster closely together.

The range of effects from variations of AR and $\sigma_{\mathrm{canting}}$ within observed limits are very different between the different
hydrometeors. Cloud ice exhibits nearly no polarimetric signals for the $\mathrm{AR_{high}}$ cases, but provides almost excessive values
compared to observations with $Z_{\mathrm{DR}} > 1\,\mathrm{dB}$ and $K_{\mathrm{DP}} > 0.1°/\mathrm{km}$ for the $\mathrm{AR_{low}} + \sigma_{\mathrm{low}}$ case, which corresponds to the B-PRO$_{\mathrm{def}}$
setting. The effect of AR and $\sigma_{\mathrm{canting}}$ variations on snow polarimetric signals is rather small. Solely, the $\mathrm{AR_{low}} + \sigma_{\mathrm{low}}$ provides
somewhat enhanced signals ($Z_{\mathrm{DR}} \approx 0.2\,\mathrm{dB}$, $K_{\mathrm{DP}} \approx 0.05°/\mathrm{km}$), but still remaining clearly below the observed polarimetric
signals ($Z_{\mathrm{DR}} \approx 0.3\,\mathrm{dB}$, $K_{\mathrm{DP}} \approx 0.1°/\mathrm{km}$) at heights where snow is expected to be the dominating hydrometeor class.

Since shape and orientation do not noticeably affect $Z_{\mathrm{H}}$, general differences between synthetic and real observations in $Z_{\mathrm{H}}$
remain. While the bright band and below-ML $Z_{\mathrm{H}}$ are matched fairly well, the FO clearly overestimates reflectivities in the
dendritic growth and aggregation layers and underestimates them in the layers where cloud ice, i.e., pristine crystals dominate.
In the upper levels, both the observed $Z_{\mathrm{DR}}$ and $K_{\mathrm{DP}}$ in the cloud ice–dominated layers fall within the range covered by the
different cloud ice shape and orientation parameterizations, with the best fit to the $\mathrm{AR_{mid}} + \sigma_{\mathrm{low}}$ and $\mathrm{AR_{low}} + \sigma_{\mathrm{high}}$ case (as far
as can be judged extrapolating by eye). However, all cloud ice paramaterizations largely overestimate $\rho_{\mathrm{hv}}$ with noticeable $\rho_{\mathrm{hv}}$
reductions only for $\mathrm{AR_{low}} + \sigma_{\mathrm{high}}$ and $\mathrm{AR_{low}} + \sigma_{\mathrm{low}}$ (the latter excluding EXP3, though). The snow-dominated lower levels (-3
to -13°C) are characterized by a strong underestimation of $Z_{\mathrm{DR}}$ and an even stronger underestimation of $K_{\mathrm{DP}}$. Only the snow's
$\mathrm{AR_{low}} + \sigma_{\mathrm{low}}$ setup raises the polarimetric signals somewhat, but it is still insufficient to get close to the observed value. The $\rho_{\mathrm{hv}}$
in lower levels also remain very close to 1 for all snow AR and $\sigma_{\mathrm{canting}}$ settings. At the heights just above the ML, where graupel
exists (i.e. in CTRL only), assuming a lower AR and especially a lower $\sigma_{\mathrm{canting}}$ can raise $Z_{\mathrm{DR}}$ and $K_{\mathrm{DP}}$ towards or beyond the
observed values. Specifically the graupel $\mathrm{AR_{mid}} + \sigma_{\mathrm{high}}$ and $\mathrm{AR_{low}} + \sigma_{\mathrm{high}}$ setups provide $Z_{\mathrm{DR}}$ that is in good agreement with the
observations, but still underestimate $K_{\mathrm{DP}}$ strongly, while $\mathrm{AR_{low}} + \sigma_{\mathrm{low}}$ provides $K_{\mathrm{DP}}$ close to the observations but also largely
overestimates $Z_{\mathrm{DR}}$.

The $Z_{\mathrm{H}}$ bright band signature agrees well between the real and synthetic observations. For the CTRL case (Fig. 11) in
particular, the absolute values of $Z_{\mathrm{H}}$ match well while the width of the layer is wider in the synthetic data. On the other hand, the
bright band is equally narrow in the observations and synthetic data for the EXP3 case (Fig. 12), but the synthetic observations
overestimate the bright band peak by about $5\,\mathrm{dBZ}$. In the polarimetric variables, generally the ML-related peak occurs slightly
below where the observations show it. For EXP3, i.e. where practically no graupel occurs, the $Z_{\mathrm{DR}}$ ML signature matches
well with the observation apart from the placement at slightly lower height. Also, the below-ML $Z_{\mathrm{DR}}$ values are in satisfactory
agreement with the observations independent of the snow AR and $\sigma_{\mathrm{canting}}$ applied. Synthetic $K_{\mathrm{DP}}$ in general underestimates the
observed value, but the snow $\mathrm{AR_{low}} + \sigma_{\mathrm{high}}$ setup provides at least a well-pronounced ML signature. An ML signature is also
seen in $\rho_{\mathrm{hv}}$, but is clearly too weak independent of the applied AR and $\sigma_{\mathrm{canting}}$ parameterizations. For the CTRL case, the ML
signatures are dominated by the polarimetric signals of graupel. The flank of increasing $Z_{\mathrm{DR}}$ with decreasing height is matched
well for setups using graupel $\sigma_{\mathrm{high}}$. However, the $Z_{\mathrm{DR}}$ increase continues to lower heights than in the observations, leading to





significantly stronger $Z_{DR}$ ML peaks ($2\,dB$ compared to observed $1\,dB$). Setups using graupel $\sigma_{low}$ even peak at $3\,dB$. The

synthetic $K_{DP}$ ML values are overestimated for $\sigma_{low}$ setups, but fit comparatively well for the $\sigma_{high}$ setups. The differences in $\rho_{hv}$ are small between the AR and $\sigma_{canting}$ settings. Despite a more pronounced $\rho_{hv}$ decrease compared to the EXP3 case, $\rho_{hv}$ remains too high compared to the observations.

Below about $1\,km$, no more differences occur in the radar variables of the different shape and orientation settings. This is because the FO's melting scheme predicts all ice hydrometeors to have melted there and having taken on the shape and

orientation properties of rain drops. These melted hydrometeors, however, still follow the size distributions of their original hydrometeor categories. That is, in the CTRL case, where the model still predicts significant amounts of fairly large-sized graupel, the FO models scattering properties of large liquid-graupel drops (with rain-like AR and $\sigma_{canting}$) that would not exist in reality, but would break up and form smaller drops. These large virtual drops dominate the polarimetric signals below the ML, causing values of $Z_{DR}$ that are far too high and values of $\rho_{hv}$ that are clearly too low.

## 5  Discussion

At model-resolvable grid scales, the difference in domain average precipitation in the sensitivity runs with reference to control runs were found to be statistically insignificant. However, the partitioning of the IWC and the source of rain drops was found to vary, with EXP3 matching qualitatively well with the retrieved hydrometeor classification from the BoXPol. The low amount of cloud ice simulated above $4\,km$ in the CTRL run is consistent with the earlier findings from Fridlind et al. (2017). This could

be compensated for the SB2M scheme by increasing $D_{ice}$ (EXP2,3), but this change was found to have negligible effects on the microphysical processes below (aggregation and riming). An increase in the temperature threshold for riming of ice and snow in the presence of supercooled raindrops independently reduced the graupel production (EXP2,3) near the melting layer. This also had no effect on the dominating aggregation process above. Hence, EXP3 is qualitatively similar to a linear combination of EXP1 and EXP2.

In the radar space, EXP3 also produced polarimetric moments much closer to the observations; i.e. the melting layer signature is better represented and the curtains of high $Z_H$ and $Z_{DR}$ values below the melting layer are prevented. However, EXP3 also produces high $Z_{DR}$ and $K_{DP}$ above $4\,km$, indicating that mid to high values of cloud ice AR would better approximate the observed polarimetric moments at this level. Regarding the excessive polarimetric signals below the ML for the CTRL run, it is possible that the FO melting scheme could be partly responsible as it virtually produces huge, physically impossible, rain-

like drops of melted graupel. However, preliminary tests with varied melting scheme setups (not shown) have not provided significantly better results. Rather, delayed melting by increasing the $T_{max}$ parameter leads to a widening or smearing out of the melting layer signatures. Other sources of uncertainty are the shape and orientation assumptions of melting graupel. Instead of a linear transition towards the description for rain drops, independent parameterizations could be developed for melting graupel, e.g., assuming more spherical and less oriented graupel particles albeit without a physical basis. The problem lies in

the prediction of significant graupel amounts of large mean sizes by the model that survive as graupel far below the expected and observed melting layer.




Both the CTRL and sensitivity runs, however, produced a low bias in the polarimetric signal at lower levels (2.5 to 4.5 km, i.e. -3 to -13°C), where snow aggregates dominate. The observed $K_{DP}$ and $Z_{DR}$ at this level are around $0.1-0.4°$/km and $0.3-0.5$ dB, while the synthetic $K_{DP}$ and $Z_{DR}$ range from $0-0.05°$/km and $0-0.1$ dB, respectively. None of the alternative shape and orientation setups for snow could provide sufficiently strong polarimetric signals to reproduce the observed signals at these heights. Snow particles could be assumed to take on even more non-spherical shapes and higher degrees of orientation to create stronger signals and tune them closer to the observed values. However, this would require going clearly outside of the reasonable and in-situ observed range of AR and $\sigma_{canting}$. Also, it is not possible to find consistent tunings that are valid for different wavelengths and observation techniques. Concerning the EMA for the ice-air mixture material of dry snowflakes, Bohren and Huffman (1983) showed that for mixtures of weakly (like ice) and non-dielectric (like air) substances, many different EMA formulas agree to first order, i.e., variations of the EMA are not expected to have a significant effect here. Applying methods like the Discrete Dipole Approximation (DDA) that solve for the scattering properties of irregularly shaped and heterogeneously structured particles, it has been demonstrated that internally homogeneous particles like (soft) spheroids, plates, or columns are not suitable proxies for fluffy, low-density hydrometeors like dendritic crystals or aggregates (Schrom and Kumjian, 2018). That is, to improve snow polarimetric signals predicted by forward operators, more realistic, less homogeneous particle models, and hence other scattering methods than the T-Matrix, need to be applied. Another cause of the low bias in the synthetic polarimetric signals at lower levels could lie in missing cloud ice that also appears sporadically as the dominant hydrometeor in the radar retrievals. The observed high $K_{DP}$ values with modest $Z_{DR}$ have also been reported for dendritic growth layer (between -10 to -20°C) for tall clouds with colder tops (-40 C) and identified to be dominated by isometric ice crystals with AR around 0.6 (Griffin et al., 2018). However, here we observe high $K_{DP}$ at a relatively warmer temperature regime (-3 to -13°C), also probably associated with isometric ice crystals (as evidenced by high $\rho_{hv}$). Such a peak in ice crystal concentration near -5°C was also reported earlier by Stewart et al. (1984) for stratiform clouds. Hence, the low bias in the model simulated synthetic polarimetric signals could be possibly due to missing additional secondary ice production (SIP) parameterizations in SB2M (e.g.,Takahashi et al. 1995; Fridlind et al. 2007; Korolev et al. 2020; Korolev and Leisner 2020; Trömel et al. 2021). Of the identified possible SIP mechanisms, ice hydrometeor collision and breakup (leading to increase in cloud ice and reduction in snow aggregates) appears to be one of the probable missing mechanism for SIP in the absence of supercooled water in the lower levels.

Finally, as discussed in the Appendix, an additional source of outstanding uncertainty not examined in these sensitivity tests is the chosen mixing formula and resultant dielectric constant. Varying the topology of the constituent phases, particularly for three-component (i.e., air–ice–water) particles within the melting layer, has been shown to have a dramatic impact on the simulated reflectivity (Fabry and Szyrmer, 1999; Battaglia et al., 2003) and $Z_{DR}$ and $K_{DP}$ (Carlin, 2018). These effects can act in concert with the aforementioned variability due to AR and $\sigma_{canting}$, and while some topologies are more physically plausible than others, it is not always clear which mixing formula best approximates the scattering properties of melting ice hydrometeors. Future work should further explore these sensitivities in an effort to constrain the degrees of freedom of the simulated polarimetric variables.





# 6  Conclusions

The model was generally found to underestimate the polarimetric signals in the lower levels (-3 to -13°C), where snow aggregates dominated the simulated hydrometeor population. Sensitivity studies with different combinations of aspect ratios and widths of each hydrometeor's canting angle distribution in the FO could also not explain this model bias, indicating 1) shortcomings in the FO and requirement of more reliable snow scattering models to draw valid conclusions from that, and/or 2) missing additional secondary ice production parameterization in the model.

In the absence of in-situ measurements using aircraft, this study shows the potential of polarimetric radar observations and radar retrievals based on high-resolution X-band radars for evaluating and improving the simulated cloud microphysical process by NWP models. For the model used in this study, improvements in the synthetic polarimetric signatures were found to be sensitive to uncertainty in the prescribed $D_{\text{ice}}$ and $T_{\text{gr}}$, while still constraining the accumulated precipitation at model resolvable scales. The latter is an important achievement for operational weather forecasting models to improve the simulated cloud microphysical processes in the model, without further degrading the forecast surface precipitation.

Future studies should make additional use of multi-frequency spectral radar polarimetric observations, which would further allow us to investigate the evolution of the ice particle size distributions (Trömel et al., 2021), along with numerical modeling to better understand the biases in modeled IWC partitioning.

*Code and data availability.* The COSMO model is distributed to research institutions free of charge under an institutional license issued by the Consortium COSMO and administered by DWD. For more information see http://www.cosmo-model.org/content/consortium/licencing.html. The COSMO license also includes access to lateral boundary data data provided by DWD. COSMO-DE analysis data used for the initial and lateral boundary conditions data for the COSMO model experiments in this study can be downloaded from the DWD database (https://www.dwd.de/DE/leistungen/pamore/pamore.html). The radar forward operator B-PRO is based on source code derived from the COSMO model, hence redistribution is limited by the COSMO license. B-PRO v2.0 used in this study is available to registered collaborators from https://git2.meteo.uni-bonn.de/git/pfo.

Modifications to the COSMO and B-PRO source codes for the sensitivity studies along with the scripts used to setup, run, and process output of COSMO and B-PRO as well as processed COSMO and B-PRO data, rain-gauge data, polarimetric radar data from BoXPol, retrievals of hydrometeor classification and scripts to produce the figures in this work are available from https://doi.org/10.5281/zenodo.5218717 (Shrestha et al., 2021).

# Appendix A:  Forward Operator Details

While focusing on polarimetry, radar FOs also require additional assumptions regarding non-polarimetric factors like the melting parameterization and the EMA applied. These are already known to impact modeled reflectivities, but can also affect the polarimetric parameters.



In B-PRO, frozen hydrometeors (i.e., cloud ice, snow, graupel, hail) are modeled as ice-air mixtures, where the air fraction is derived from the model-provided size-mass relation and the hydrometeor shape assumptions applied in the FO. B-PRO includes a melting model: above a given, hydrometeor class–specific temperature $T_{\mathrm{meltbegin}}$, ice hydrometeors start to melt and turn into water-ice-air mixtures. For the modeling of the effective refractive index of the ice-air and water-ice-air mixture

particles, a selection of effective medium approximations (EMA) are available, namely 2- and 3-component Maxwell Garnett, Bruggemann, and Debye mixing rules which are popular in the meteorological radar community (see Blahak, 2016). Note that different EMAs have been theoretically derived under widely different assumptions and for different scattering phenomena (Bohren and Huffman, 1983) and lead to a widespread range of possible solutions for radar signals for the same hydrometeors (see Blahak, 2016, for effects on reflectivity). Because it is often unclear which EMA "best" suits which situation and which

radar parameter(s), it should perhaps be treated in the sense of an ensemble based on many different EMA choices.

As explained in Sect. 2.2, the COSMO model and its SB2M scheme instantaneously transfer meltwater from the ice hydrometeor classes into rain. Taken literally, that implies that ice class hydrometeors are always (completely) frozen, and no mixed-phase hydrometeors exist. When only assuming pure phase hydrometeors, however, radar forward operators have issues producing realistic melting layer signatures. Some FOs try to compensate for the instantaneous meltwater shedding by artifi-

cially redistributing the rain present in a grid box back to the melting hydrometeors, assuming that all of the rain in the grid box comes from melting and no shedding occurs until melting is complete (Wolfensberger and Berne, 2018; Jung et al., 2008). While being an attractive and modern concept, this might systematically overestimate the mean sizes of melting particles and lead to artificial discontinuities of the polarimetric parameters at the lower edge of the melting layer (Wolfensberger and Berne, 2018). B-PRO instead assumes a mass fraction of the ice hydrometeors to be melted, i.e. turns a part of what the model predicts

as unmelted ice into (unshedded) liquid water. In contrast to the redistribution approach above, this ensures continuity of the radar signals over the melting layer edges but might lead to an underestimation of the mean sizes, and hence of the radar signals, of the melting hydrometeors. For practical purposes, a possible lack of "bright-band" may be artificially compensated by choosing an EMA from the available options which produces stronger melting signatures. The B-PRO melting model, inherited from EMVORADO (see Blahak, 2016), predicts a temperature- and particle size-dependent meltwater mass fraction for the ice

hydrometeor classes once the ambient temperature exceeds a specified class threshold $T_{\mathrm{meltbegin}}$. The melt fraction decreases with increasing $D$ for constant $T$ and grows exponentially with $T$ at constant $D$. All particles of a given hydrometeor class are considered completely melted once a given temperature $T_{\mathrm{max}}$ is reached. By default, the melting scheme is dynamic, i.e., $T_{\mathrm{max}}$ is within specifiable limits determined from the model temperature and hydrometeor fields in each model column.

Hydrometeor scattering properties are calculated for single particles over a range of sizes per hydrometeor class, applying the

size-dependent shape and melting fraction values. Bulk scattering properties per hydrometeor class are obtained by integrating the monodisperse scattering properties over the particle size distribution provided by COSMO applying the Simpson quadrature rule.

The temperature thresholds governing the degree of melting of the particles as well as the EMA for each hydrometeor type can be controlled by the user through a (FORTRAN) namelist file. For this study, we defined a FO setup as a baseline, B-

PRO$_{\mathrm{def}}$, using reflectivity observations from BoXPol as well as synthetic reflectivities from EMVORADO (Zeng et al., 2016)



as reference. The B-PRO default melting scheme parameters (see Blahak, 2016) are more suitable for convective situations. For example, the default $T_{\mathrm{meltbegin}}$ for graupel and hail are -10°C to reflect the effects of wet growth in convective updrafts, but would not be suitable for stratiform clouds. Hence, for the study baseline setup in the present study the melting model parameters have been adapted for stratiform situations. $T_{\mathrm{meltbegin}}$ for all hydrometeors is set to 0°C. Also, the dynamic melting

fraction scheme has been switched off, and $T_{\mathrm{max}}$ is replaced by a fixed, hydrometeor-class specific value, except for snow. Snow applies the dynamic scheme, but in the case studied here it practically behaves as if $T_{\mathrm{max}}^{\mathrm{snow}}$ was set to 3°C. In addition, the EMA of dry and wet snow, which are modeled as a homogeneous single-layer spheroid in B-PRO but as a 2-layered sphere in EMVORADO, has been chosen to roughly reproduce the melting snow reflectivities as predicted by EMVORADO. All settings are documented in Table 2.

*Author contributions.*   PS, JM, ST, UB designed the study, carried out the analysis and together wrote the manuscript. ST conceptualized the case study and set up the QVP radar data analysis and processing. PS conducted the model simulations, FO runs and QVP processing of the model data. JM made adaptations to the FO and designed and conducted the FO sensitivity runs. UB aided in the model and FO sensitivity runs. VP provided the radar retrievals for hydrometeor classification. JC aided in updating the polarimetry physics in the FO.

*Competing interests.*   The authors declare that they have no conflict of interest.

*Acknowledgements.*   The research was carried out in the framework of the Priority Programme SPP-451 2115 "Polarimetric Radar Observations meet Atmospheric Modelling (PROM)" funded by the German Research Foundation (DFG). Prabhakar Shrestha acknowledges his support for PROM sub-project ILACPR (Grant SH 1326/1-1). J. Mendrok and V. Pejcic carried out their work under PROM sub-project Operation Hydrometeors (Grants BL 945/2-1 and TR 1023/16-1). Funding for J. T. Carlin was provided by NOAA/Office of Oceanic and Atmospheric Research under NOAA-University of Oklahoma Cooperative Agreement #NA16OAR4320115, U.S. Department of Commerce.
We gratefully acknowledge the computing time (project HBN33) granted by the John von Neumann Institute for Computing (NIC) and provided on the supercomputer JUWELS at Jülich Supercomputing Centre (JSC). The post-processing of model output data and input/output for FO was done using the NCAR Command language (Version 6.4.0). We would like to thank the city of Bonn for providing the rain gauge data. We also acknowledge the support of Kai Mühlbauer and the open source radar library wradlib (https://docs.wradlib.org/en/stable/index.html) regarding the processing of radar data and the optimization of codes.





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



**Figure 1.** Time series of quasi-vertical profiles measured with the polarimetric X-band radar in Bonn (BoXPol) on 16 Nov 2014. Panels show (a) horizontal reflectivity $Z_\mathrm{H}$, (b) differential reflectivity $Z_\mathrm{DR}$, (c) specific differential phase $K_\mathrm{DP}$, and (d) cross-correlation coefficient $\rho_\mathrm{hv}$. Overlaid solid black contours depict the $Z_\mathrm{H}$ field in 5-dBZ increments, while black dashed lines indicate the COSMO isotherms for the radar location.



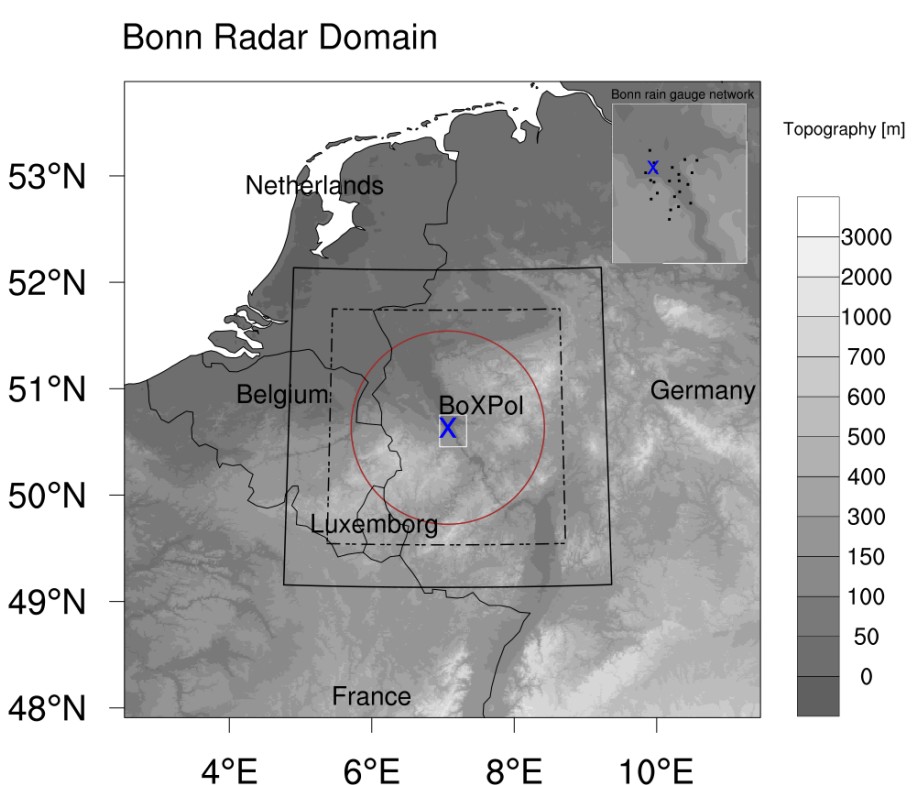

**Figure 2.** Topography of the model domain showing the Rhine Massif, the Rhine Valley and the northwestern lowlands. The dash-dotted box indicates the inner boundary (excluding the relaxation zone) used to compute domain average precipitation. The location of the X-band polarimetric radar at Bonn (BoXPol) and the radial extent of its observations are shown as a blue X and red circle, respectively. The white box around the BoXPol location indicates the location of the Bonn rain gauge network, and the inset map shows a zoomed in view of the network area along with the locations of the rain gauges (black dots).



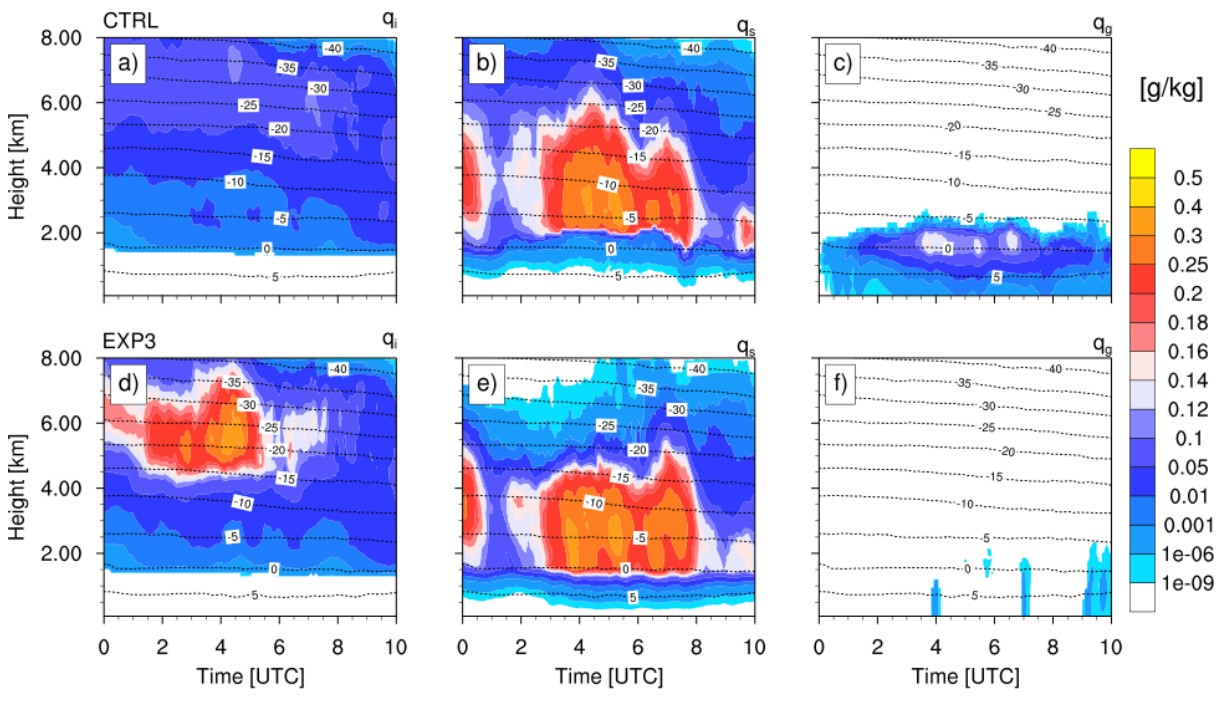

**Figure 3.** QVPs of model predicted hydrometeor mixing ratios of cloud ice, snow, and graupel. Overlaid dashed lines are contours of modeled air temperature QVPs.

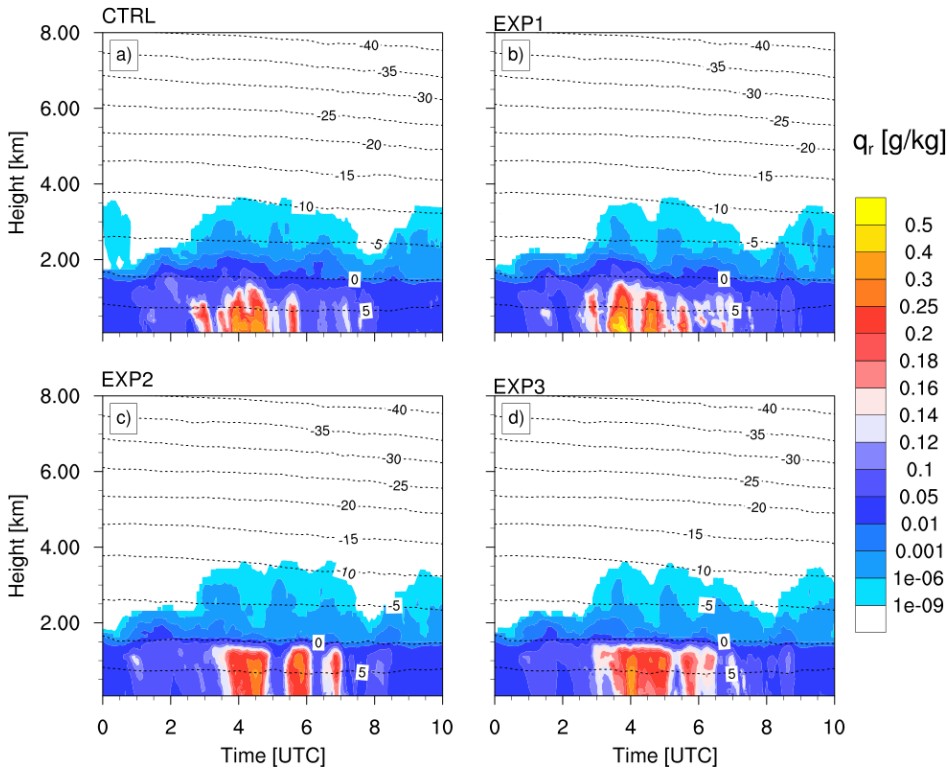

**Figure 4.** QVPs of modeled rain mixing ratio from the different model experiment runs. Dashed lines are contours of modeled air temperature QVPs.





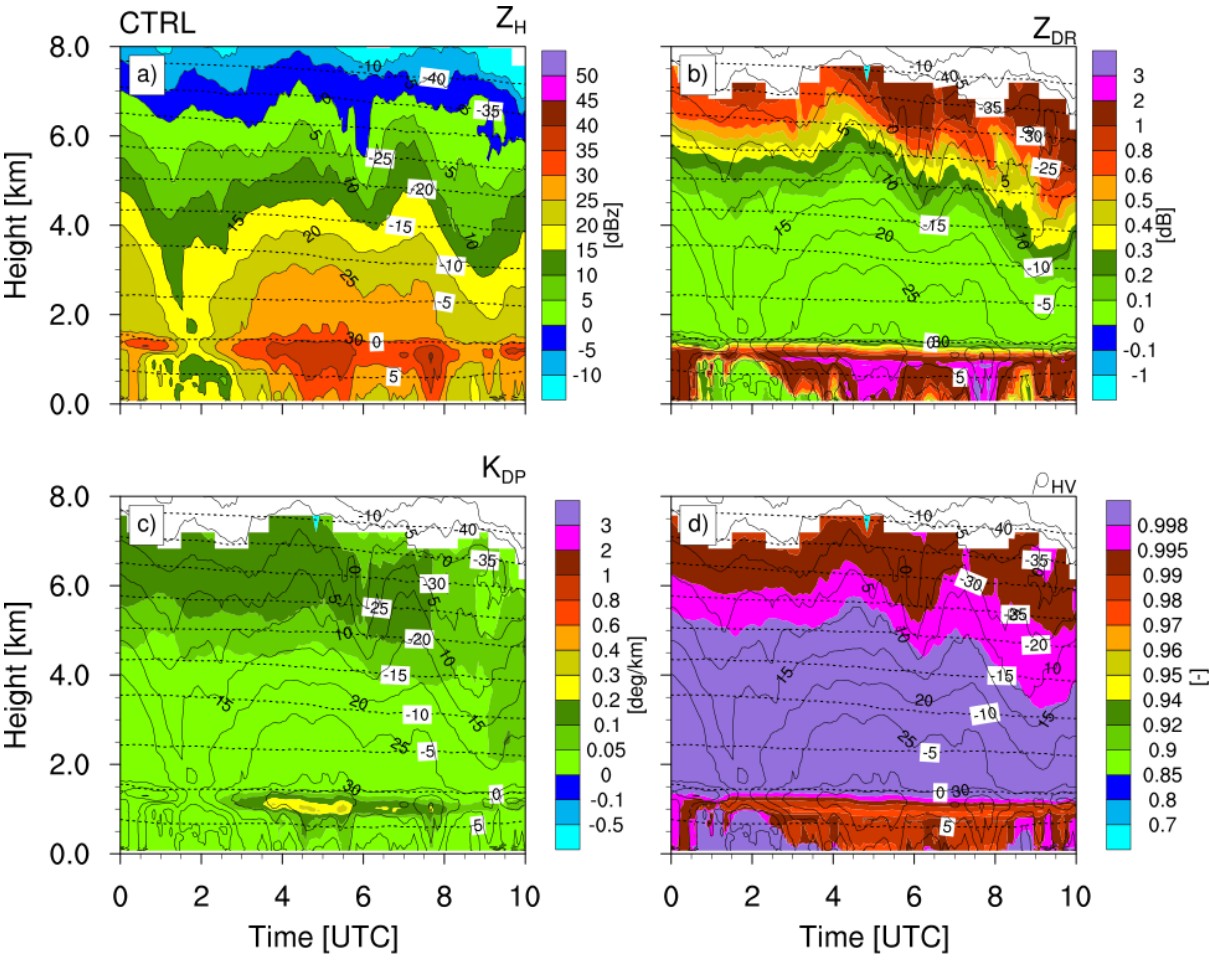

**Figure 5.** Synthetic QVPs of horizontal reflectivity $Z_H$, differential reflectivity $Z_{DR}$, specific differential phase $K_{DP}$, and cross-correlation coefficient $\rho_{hv}$ for the CTRL run and applying the B-PRO$_{def}$ setup. Radar variable color scales are identical to the ones applied in Fig. 1. Also shown are contours of air temperature.



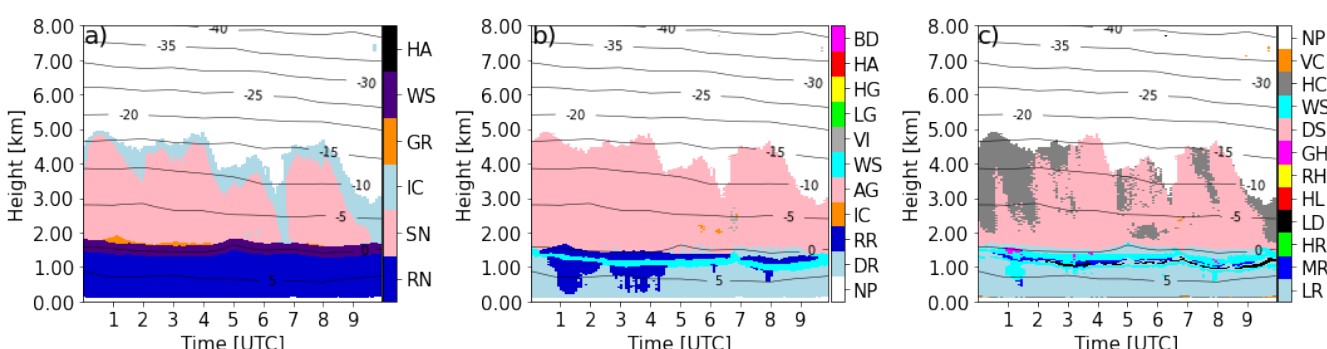

**Figure 6.** Time series of dominant hydrometeor types retrieved from the observed QVPs. a) HCA-Pejcic with rain (RN), snow (SN), cloud ice (IC), graupel (GR), wet snow (WS) and hail (HA). b) HCA-Dolan with no precipitation (NP), drizzle (DR), rain (RR), ice crystals (IC), aggregates (AG), wet snow (WS), vertical aligned ice (VI), low density graupel (LG), high density graupel (HG), hail (HA), big drops/melting snow (BD). c) HCA-Zrnic with light rain (LR), moderate rain (MR), heavy rain (HR), large drops (LD), hail (HL), rain/hail (RH), graupel/hail (GH), dry snow (DS), wet snow (WS), horizontal ice (HC), vertical ice (VI) and no precipitation (NP).



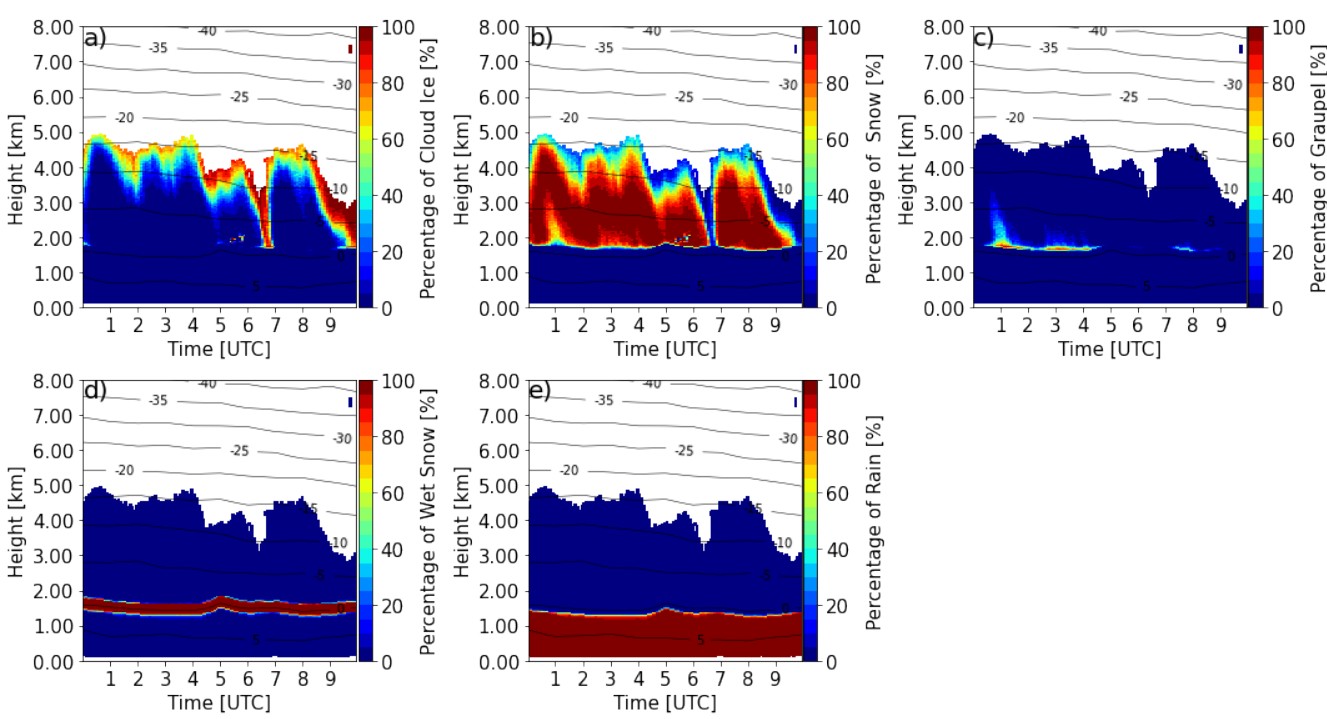

**Figure 7.** Time series of hydrometeor percentage retrieved from the observed QVPs with HCA-Pejcic for cloud ice (a), snow (b), graupel (c), wet snow (d) and rain (e).





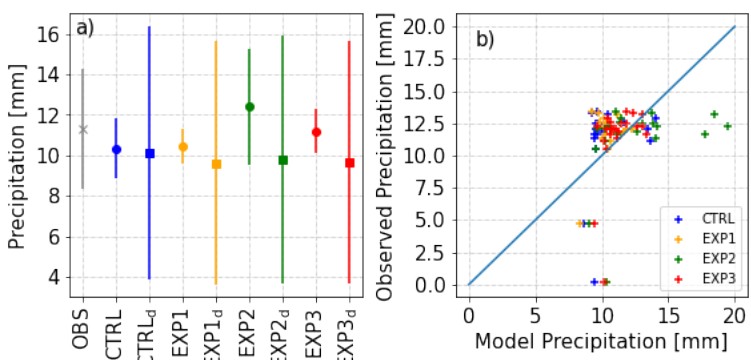

**Figure 8.** (a) Comparison of mean accumulated precipitation (00:00 UTC – 10:00 UTC) as measured by 22 rain gauges around Bonn (cross), as predicted by COSMO at the model grid points corresponding to the 22 gauge locations (circles) and over the inner model domain (squares) for different model setups. The vertical bars indicate the standard deviation. (b) Scatter plot of precipitation totals between observations and model for each rain gauges. Different colors are used for the multiple experiments.





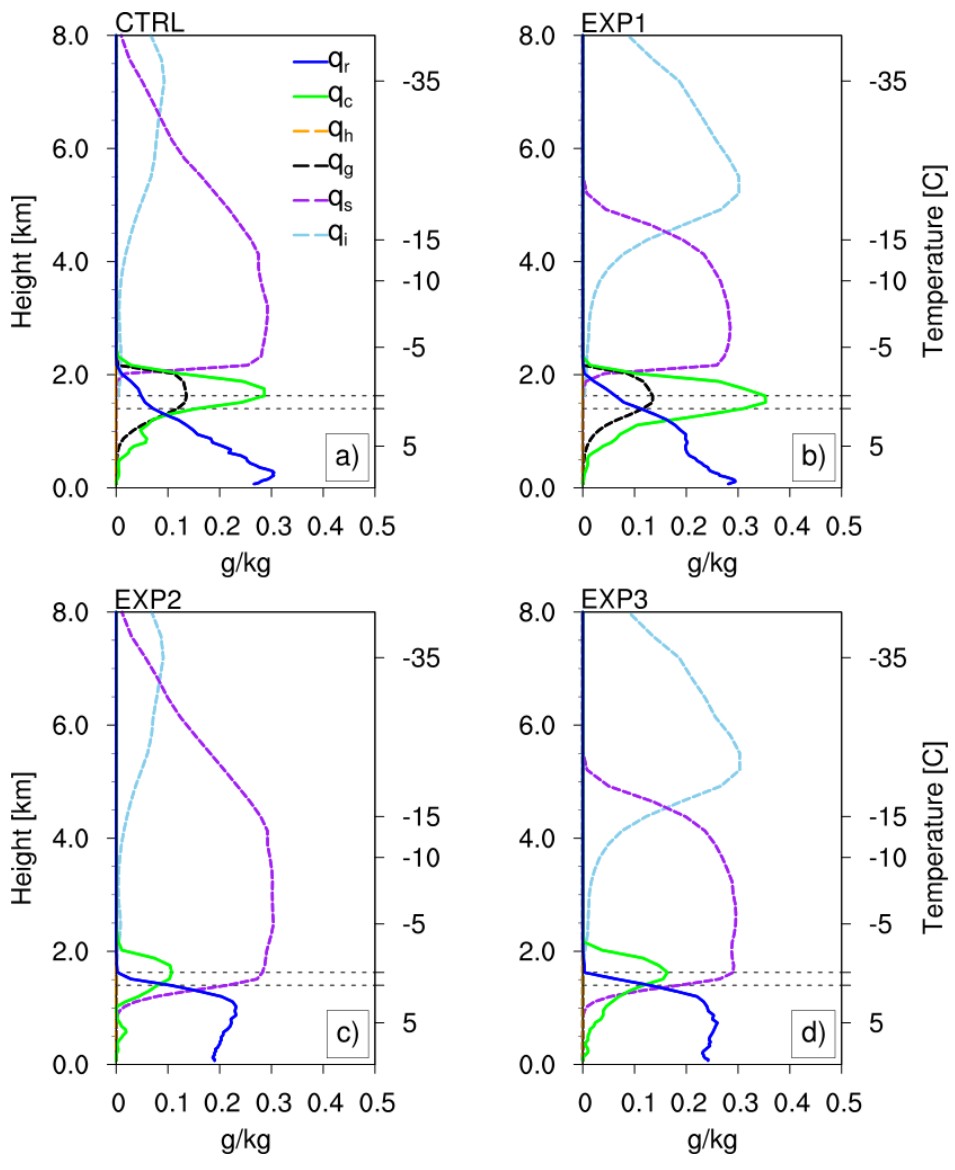

**Figure 9.** Half-hourly averaged QVPs of modeled hydrometeor mixing ratio around 04:00UTC on 16 Nov 2014. On the right axis, air temperatures corresponding to the heights are denoted. Dashed gray lines indicate temperature thresholds of $\pm\,0.5°C$ in the melting layer.





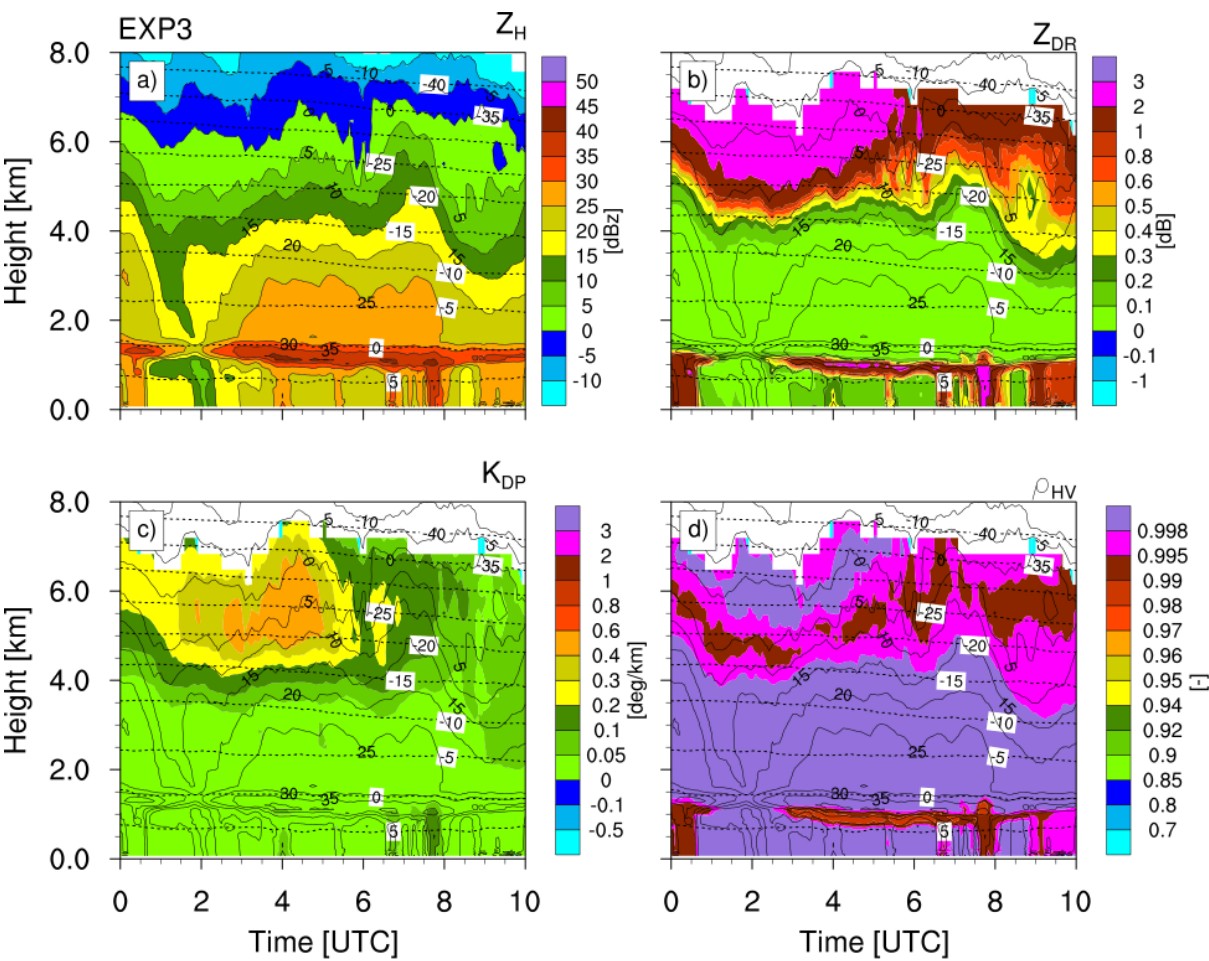

**Figure 10.** As Fig. 5, but for model experiment setup EXP3.





**Figure 11.** Median synthetic profiles of (left) $Z_H$, (middle-left) $Z_{DR}$, (middle-right) $K_{DP}$, and (right) $\rho_{hv}$ with different shape and orientation assumptions for (top) cloud ice, (middle) snow, and (bottom) graupel hydrometeor classes of the CTRL run. See Table 5 for the specific shape and orientation parameterizations used. Thick lines denote results of the sensitivity runs for the B-PRO$_{def}$ setup. Equivalent median QVP from the full experiment runs (grey lines) and observations (black lines) are added as reference.



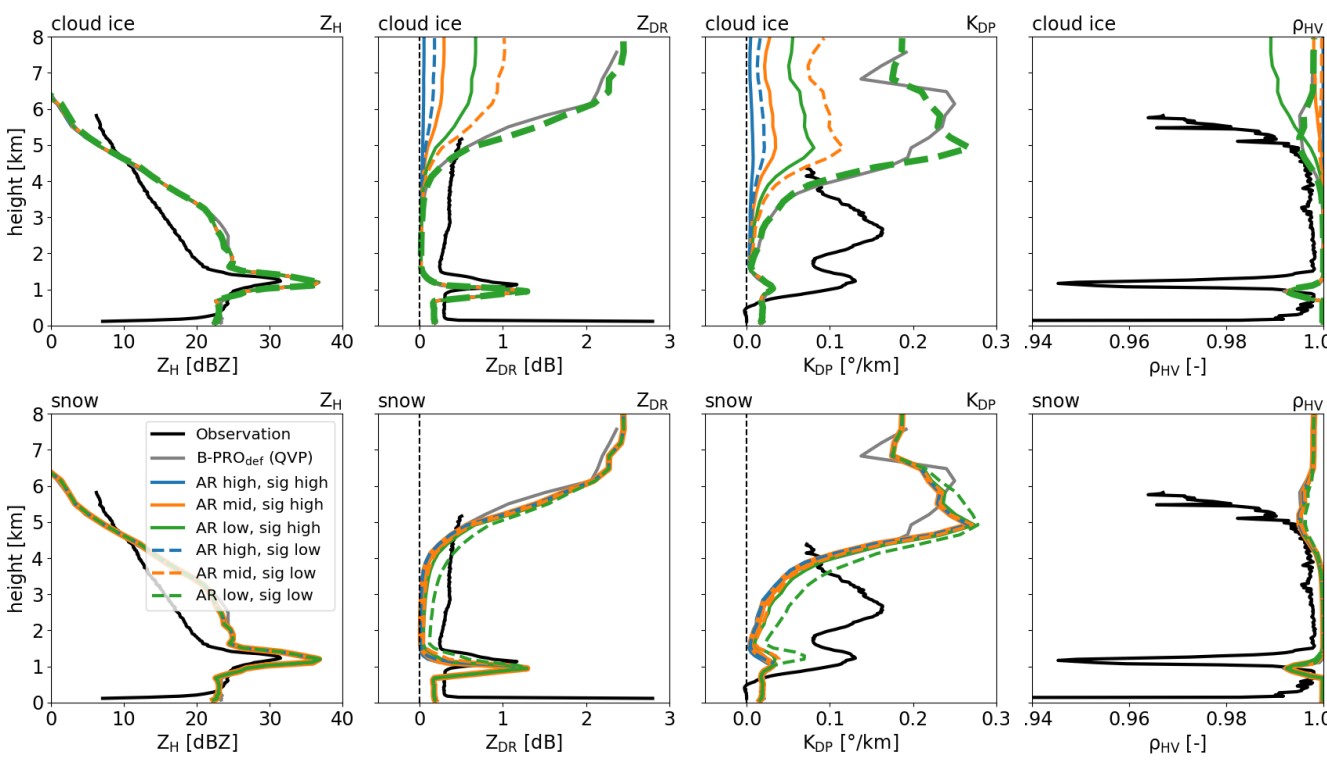

**Figure 12.** As Fig. 11, but for model experiment EXP3. No graupel panel is shown here, since graupel mixing ratios in this experiment are too low to produce noticeable contributions to the radar signal.



**Table 1.** Parameters of the size-mass and velocity-mass relationships following Eq. (2) and Eq. (3) used in the SB2M. These refer to $D$ in units of m, $x$ in kg, and $v_T$ in $\mathrm{m\,s^{-3}}$. The last two columns contain the shape parameters of the assumed mass distribution. $D_{x,min} = a_g x_{min}^{b_g}$ and $D_{x,max} = a_g x_{max}^{b_g}$ are the diameters corresponding to the mass limits $x_{min}, x_{max}$ and are added for better interpretation.

| | $\mathbf{a_{geo}}$ | $\mathbf{b_{geo}}$ | $\mathbf{a_v}$ | $\mathbf{b_v}$ | $\mathbf{x_{min}}$ | $\mathbf{x_{max}}$ | $\mathbf{D_{x,min}}$ | $\mathbf{D_{x,max}}$ | $\mu$ | $\nu$ |
|---|---|---|---|---|---|---|---|---|---|---|
| **Cloud Liquid** | 0.124 | 1/3 | $3.75 \cdot 10^5$ | 2/3 | $4.2 \cdot 10^{-15}$ | $2.6 \cdot 10^{-10}$ | $2.0 \cdot 10^{-6}$ | $8.0 \cdot 10^{-5}$ | 0 | 1/3 |
| **Rain** | 0.124 | 1/3 | 114.0 | 0.234 | $2.6 \cdot 10^{-10}$ | $3.0 \cdot 10^{-6}$ | $8.0 \cdot 10^{-5}$ | $1.8 \cdot 10^{-3}$ | 0 | 1/3 |
| **Cloud Ice** | 0.835 | 0.390 | 27.7 | 0.216 | $1.0 \cdot 10^{-12}$ | $1.0 \cdot 10^{-6}$ | $1.7 \cdot 10^{-5}$ | $1.6 \cdot 10^{-3}$ | 0 | 1/3 |
| **Snow** | 2.4 | 0.455 | 4.2 | 0.092 | $1.0 \cdot 10^{-10}$ | $2.0 \cdot 10^{-5}$ | $6.8 \cdot 10^{-5}$ | $1.8 \cdot 10^{-2}$ | 0 | 1/2 |
| **Graupel** | 0.142 | 0.314 | 86.89 | 0.268 | $1.0 \cdot 10^{-9}$ | $5.0 \cdot 10^{-4}$ | $2.1 \cdot 10^{-4}$ | $1.3 \cdot 10^{-2}$ | 1 | 1/3 |
| **Hail** | 0.1366 | 1/3 | 39.3 | 1/6 | $2.6 \cdot 10^{-9}$ | $5.0 \cdot 10^{-4}$ | $1.9 \cdot 10^{-4}$ | $1.1 \cdot 10^{-2}$ | 1 | 1/3 |





**Table 2.** Overview of B-PRO settings used as the baseline setup B-PRO$_{\text{def}}$ in this study. This includes both hard-coded internal parameters (marked by *) as well as parameters controllable by the user via a namelist file. For the dynamic melting schemes, $T_{\text{max}}$ gives the lower and upper bounds that $T_{\text{max}}$ is permitted to exhibit. $q_{\text{min}}$ and $n_{\text{min}}$ are applicable in cases of the dynamic melting scheme only, giving the lower limits of specific mass and number density of the respective hydrometeor category at a model grid point, respectively, to perform a $T_{\text{max}}$ update. For the meaning of the EMA tags see Blahak (2016).

| Parameter | cloud ice | snow | graupel | hail |
|---|---|---|---|---|
| Melting scheme | | | | |
| $T_{\text{meltbegin}}$ [°C] | 0.0 | 0.0 | 0.0 | 0.0 |
| $T_{\text{min}}$ [°C] | 0.0 | 0.0 | 0.0 | 0.0 |
| $T_{\text{max}}$ type | fixed | dynamic | fixed | fixed |
| $T_{\text{max}}$ [°C] | 5.0 | $3.0-10.0$ | 5.0 | 20.0 |
| $q_{\text{min}}, n_{\text{min}}$ | − | $10^{-8}, 10^{0}$ | − | − |
| EMA | | | | |
| dry | mas | mas | mis | mis |
| wet | mawsms | mawsms | mawsms | mawsms |
| PSD integration* | | | | |
| $D_{\text{min}}, D_{\text{max}}$ [mm] | 0.02, 4.0 | 0.05, 30.0 | 0.01, 30.0 | 0.05, 100.0 |
| Microphysics | | | | |
| AR | $\approx 0.2$ | $max(0.7-10D, 0.5)$ | $max(1.0-20D, 0.8)$ | $max(1.0-20D, 0.8)$ |
| | Andrić et al. (2013), plates | Xie et al. (2016) | Ryzhkov et al. (2011) | Ryzhkov et al. (2011) |
| $\sigma_{\text{canting}}$ [°] | 12.0 | 40.0 | 40.0 | 40.0 |
| | Matrosov et al. (2005) | Ryzhkov et al. (2011) | Ryzhkov et al. (2011) | Ryzhkov et al. (2011) |



**Table 3.** List of COSMO model simulations with perturbed parameters to account for uncertainty in the partitioning of the ice water content.

| Description | $D_{ice}$ | $T_{gr}$ |
|---|---|---|
| | [μm] | [K] |
| CTRL (default run) | 50.0 | 273.2 |
| EXP1 | 400.0 | 273.2 |
| EXP2 | 5.0 | 268.2 |
| EXP3 | 400.0 | 270.2 |





**Table 4.** Student's t-test for differences in domain average precipitation with reference to CTRL run. Calculated t-statistic and the p-values for horizontal resolutions $\Delta x$ of 1.1 km and 11 km.

| Exp | $\Delta x = 1.1\,\mathrm{km}$ | | $\Delta x = 11.0\,\mathrm{km}$ | |
| --- | --- | --- | --- | --- |
| | t-stat | p-values | t-stat | p-values |
| EXP1 | 12.9 | $2.8 \cdot 10^{-38}$ | 1.3 | 0.2 |
| EXP2 | 8.3 | $8.6 \cdot 10^{-17}$ | 0.8 | 0.4 |
| EXP3 | 12.0 | $6.6 \cdot 10^{-33}$ | 1.2 | 0.2 |

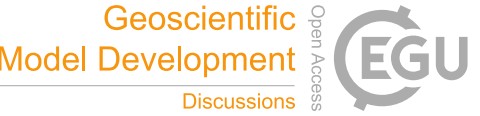

**Table 5.** Overview of settings for the FO sensitivity study. The first line of each entry gives the source and the second the parameterization or value. Here, $D$ is in terms of maximum diameter [m]. Entries typeset in bold denote the setup corresponding to B-PRO$_{def}$ for the respective hydrometeor class. Cloud ice AR$_{low}$ from Andrić et al. (2013) applies their density-size parameterization for plates calculating AR following COSMO's m-D relation from the resulting volume. Setting $min(\text{AR}) = 0.2$ in order to ensure a stable T-matrix solution leads to AR $= 0.2$ for all but the smallest crystals (also, the AR parameterizations for plates and dendrites are practically identical in this range).

|  | cloud ice | snow | graupel |
|---|---|---|---|
| AR$_{high}$ | Xie et al. (2016) $max(0.9 - 10^3 D, 0.7)$ | Ryzhkov et al. (2011) $max(1.0 - 20D, 0.8)$ | **Ryzhkov et al. (2011)** $\boldsymbol{max(1.0 - 20D, 0.8)}$ |
| AR$_{mid}$ | Matsui et al. (2019) 0.35 | **Xie et al. (2016)** $\boldsymbol{max(0.7 - 10D, 0.5)}$ | Putnam et al. (2017) 0.75 |
| AR$_{low}$ | **Andrić et al. (2013)** $\boldsymbol{\approx 0.2}$ | Dunnavan et al. (2019) 0.4 | Straka et al. (2000) 0.6 |
| $\sigma_{high}$ | Xie et al. (2016) 40° | **Ryzhkov et al. (2011)** **40°** | **Ryzhkov et al. (2011)** **40°** |
| $\sigma_{low}$ | **Carlin (pers. comm., 2020)** **12°** | Matsui et al. (2019) 20° | Putnam et al. (2017) 10° |