# Peer review of "Evaluation of the COSMO model (v5.1) in polarimetric radar space – Impact of uncertainties in model microphysics, retrievals, and forward operator"

_Geoscientific Model Development, 2021_

## Referee Comment (RC1)

Review GMD-2021-188

Title: Evaluation of the COSMO model (v5.1) in polarimetric radar space – Impact of uncertainties in model microphysics, retrievals, and forward operator

Authors: Prabhakar Shrestha, Jana Mendrok, Velibor Pejcic, Silke Trömel, Ulrich Blahak, and Jacob T. Carlin

Recommendation: Accept with minor revisions

General comments:

This paper presents an evaluation of the COSMO model for a stratiform precipitation event over Germany. The evaluation is performed using a polarimetric radar network and rain gauges. On the one hand, the evaluation is done using a model-to-observations approach, retrieving synthetic polarimetric signatures from the model with the Bonn Polarimetric Radar Forward Operator. This is complemented by an observations-to-model approach, retrieving synthetic model fields from the observations using several Hydrometeor Classification Algorithms.

The paper discusses a number of fairly simple, but relevant sensitivity tests, including two conversion thresholds within the model microphysics parameterization, and aspect ratio and canting angle assumptions within the forward operator.

Using the model-to-observations approach in combination with an observations-to-model approach, the authors demonstrate nicely which aspects of the evaluation point to real issues with the model assumptions (e.g. overprediction and too slow melting of graupel particles in the default model near and below the melting level, as well as an overprediction of large snow aggregates aloft). Issues with the forward operator are also highlighted as the too large cross-correlation coefficient in all experiments suggest a lack of variability in shapes of the ice hydrometeors.

While the experiments are fairly simple, I think the paper is well-written and structured and presents a very nice example of state-of-the-art techniques in model evaluation with relevant recommendations to the scientific community. Hence, I only have a few minor comments that should probably be addressed before I would recommend acceptance for publication in *Geoscientific Model Development*.

Minor comments:

- L25: Maybe it is worth mentioning the P3 scheme here (Morrison and Milbrandt, 2015), as an example of a microphysics scheme that no longer requires a hard separation in hydrometeor categories.
- L115: Since the authors are discussing *size* distributions here, shouldn't this be the *third* and *zeroth* moment, rather than the *zeroth* and *first* moments respectively?
- L125: small typo: …, it**s** mu depends on….
- L230: Do the authors mean a 340 km by 340 km domain, rathe rather than a domain of 340 $km^2$? If the latter, that would be a very small domain…
- Figure 3: It is worth indicating explicitly in the caption that panel a refers to cloud ice, panel b to snow etc..
- L312: It is worth referring to Figure 3 here for comparison against the model.

- Table 3: One possibly larger comment is about the microphysics experiment design. How did the authors pick the different values for the snow auto-conversion threshold and the graupel temperature threshold? More specifically, I am not sure I understand the rationale for the differences between EXP2 and EXP3. Wouldn't it be cleaner to only vary the $T_{graupel}$ in EXP2 and use values of $D_{ice}$ = 50 μm and $T_{graupel}$ = 270.2 K? At the very least, it is not clear to me why the $T_{graupel}$ is different between EXP2 and EXP3? Since EXP2 is hardly mentioned, I feel that it might even be worth just removing the experiment from the table and all discussions altogether.
- L340: Not sure I agree that the qr between CTRL and EXP1 are similar. There appear to be much larger peak values of qr in EXP1 than in the CTRL.
- Figure 4: Could the authors add the panels for EXP1 as well here? That would show more clearly the impact of only the $D_{ice}$ change.
- L400: Compare against? Do the authors mean compare Figure 11 against Figure 5?
- L430: Could the authors speculate as to why the $AR_{low}$ + $\sigma_{low}$ could lead to a reduction in $\rho_{HV}$? I would think that a low aspect ratio and low canting angle would lead to more uniform behaviour and hence a larger $\rho_{HV}$.

References:

Morrison, H., J.A Milbrandt, 2015: Parameterization of Cloud Microphysics Based on the Prediction of Bulk Ice Particle Properties. Part I: Scheme Description and Idealized Tests, Journal of the Atmospheric Sciences, 72, 287-311.

---

## Author Comment (AC1)

Review GMD-2021-188

Title: Evaluation of the COSMO model (v5.1) in polarimetric radar space – Impact of uncertainties in model microphysics, retrievals, and forward operator

Authors: Prabhakar Shrestha, Jana Mendrok, Velibor Pejcic, Silke Trömel, Ulrich Blahak, and Jacob T. Carlin

Recommendation: Accept with minor revisions

General comments:

This paper presents an evaluation of the COSMO model for a stratiform precipitation event over Germany. The evaluation is performed using a polarimetric radar network and rain gauges. On the one hand, the evaluation is done using a model-to-observations approach, retrieving synthetic polarimetric signatures from the model with the Bonn Polarimetric Radar Forward Operator. This is complemented by an observations-to-model approach, retrieving synthetic model fields from the observations using several Hydrometeor Classification Algorithms.

The paper discusses a number of fairly simple, but relevant sensitivity tests, including two conversion thresholds within the model microphysics parameterization, and aspect ratio and canting angle assumptions within the forward operator.

Using the model-to-observations approach in combination with an observations-to-model approach, the authors demonstrate nicely which aspects of the evaluation point to real issues with the model assumptions (e.g. overprediction and too slow melting of graupel particles in the default model near and below the melting level, as well as an overprediction of large snow aggregates aloft). Issues with the forward operator are also highlighted as the too large cross-correlation coefficient in all experiments suggest a lack of variability in shapes of the ice hydrometeors.

While the experiments are fairly simple, I think the paper is well-written and structured and presents a very nice example of state-of-the-art techniques in model evaluation with relevant recommendations to the scientific community. Hence, I only have a few minor comments that should probably be addressed before I would recommend acceptance for publication in Geoscientific Model Development.

**We are very thankful for the reviewer's comments. Below, we address the reviewer's specific comments (in bold blue).**

Minor comments:

- L25: Maybe it is worth mentioning the P3 scheme here (Morrison and Milbrandt, 2015), as an example of a microphysics scheme that no longer requires a hard separation in hydrometeor categories.

**Yes, good idea. We added the reference and included the statement**

**Ln 26: "Morrison and Milbrandt (2015) developed an alternative scheme called P3 with only a single frozen hydrometeor class but with explicit prediction of size-dependent hydrometeor bulk densities and fall speeds, based on the prognostic rimed and deposited masses. Such schemes are often tuned in NWP models to …"**

- L115: Since the authors are discussing size distributions here, shouldn't this be the third and zeroth moment, rather than the zeroth and first moments respectively?

**Yes, this formulation was indeed confusing. Zeroth and the first moment are correct, because it is a particle mass distribution (PMD) rather than a PSD. We added a clarifying remark and included the transformation formula from PMD to PSD after Eq. (3).**

- L125: small typo: …, its mu depends on….

**Fixed.**

- L230: Do the authors mean a 340 km by 340 km domain, rathe rather than a domain of 340 km2? If the latter, that would be a very small domain…

**Yes, it is a 340 km x 340 km domain. Fixed.**

- Figure 3: It is worth indicating explicitly in the caption that panel a refers to cloud ice, panel b to snow etc..

**The Figure 3 caption has been updated in the revised manuscript.**

**"QVPs of the model predicted hydrometeor mixing ratios of cloud ice (a,d,g), snow (b,e,h), and graupel (c,f,i) for the CTRL (top row), EXP1 (middle row) and EXP3 (bottom row) runs. Overlaid dashed lines are contours of modeled air temperature QVPs."**

- L312: It is worth referring to Figure 3 here for comparison against the model.

**The reference to Fig. 3 has been added.**

- Table 3: One possibly larger comment is about the microphysics experiment design. How did the authors pick the different values for the snow auto-conversion threshold and the graupel temperature threshold? More specifically, I am not sure I understand the rationale for the differences between EXP2 and EXP3. Wouldn't it be cleaner to only vary the Tgraupel in EXP2 and use values of Dice = 50 μm and Tgraupel = 270.2 K? At the very least, it is not clear to me why the Tgraupel is different between EXP2 and EXP3? Since EXP2 is hardly mentioned, I feel that it might even be worth just removing the experiment from the table and all discussions altogether.

**For the snow auto-conversion threshold, we conducted a sensitivity study using multiple values of Dice (e.g., 5, 50, 150, 400, 800 $\mu m$), the default value being 50 $\mu m$. From these experiments, Dice = 400 $\mu m$ showed the best improvement in the synthetic polarimetric signatures at upper levels, and is used in this study. With Dice=800 $\mu m$, snow-to-ice conversion is limited too much. We chose both the lower and the upper margins of the Dice experimentation range to also check if it has any effect on surface precipitation.**

**We varied Tgr from the default 0°C by reducing it by 5 and 3 °C for EXP2 and EXP3 respectively, to check the sensitivity of graupel production near the melting layer. It showed that the threshold of -3°C was already adequate to reduce the apparently spurious graupel production and that further reduction to -5°C is not necessary. Of course one wants to be as conservative with such thresholds as possible to not affect other, e.g. convection, types of clouds. This also had no effect on the aggregation process above as the change in Dice has negligible effect on this process. Results from EXP2 are discussed in section 4.1.2, with figures in Fig.4 and 9.**

**EXP 1,2,3,4 exhibit different combinations of aggregation (ice/snow partitioning) and riming (graupel production and rain gradient below melting layer), while producing similar domain average precipitation. So, we think all the experiments and discussion therein supports the study.**

**The following paragraph has been added in the revised manuscript to clarify the setup of model sensitivity study:**

**Ln 335: "For the cloud ice aggregation threshold, we conducted a sensitivity study using multiple values of Dice (e.g., 5, 50, 150, 400, 800 $\mu m$), the default value being 50 $\mu m$. For brevity, we only report on the results from one lower and one upper value as well as the default value. From these experiments, Dice = 400 $\mu m$ showed the best improvement in the synthetic polarimetric signatures and is used as the upper Dice value in this study. Similarly, we varied Tgr from the default 0°C by reducing it by 5 and 3 °C respectively, to check the sensitivity**

of graupel production near the melting layer. The four experiments together constitute different combinations of aggregation (ice/snow partitioning) and riming (graupel production and rain gradient below melting layer).”

- L340: Not sure I agree that the qr between CTRL and EXP1 are similar. There appear to be much larger peak values of qr in EXP1 than in the CTRL.

**Here, we are suggesting that CTRL and EXP1 do not show the sharp gradient in "qr" as observed for EXP2 and EXP3. However, they do differ in terms of peak values as suggested by the reviewer. The sentence has been rephrased in the revised manuscript:**

**Ln 351: "For example, CTRL and EXP1 do not show the sharp gradient in qr near the melting layer as simulated for EXP2 and EXP3 (Fig. 4). For CTRL and EXP1, qr increases gradually below the melting layer, but differ in peak values. "**

- Figure 4: Could the authors add the panels for EXP1 as well here? That would show more clearly the impact of only the Dice change.

**We assume that Figure 3 instead of 4 is meant here, because in Figure 4 there is already a panel for EXP1. We have now added EXP1 in the revised Figure 3 and the text in Section 4.1.2 has also been modified accordingly.**

[Figure]

- L400: Compare against? Do the authors mean compare Figure 11 against Figure 5?

**Fig.11 (and also Fig.12) includes the median profiles from FO runs with the baseline setup from both the full domain QVP (grey line) and the single-column profiles (thick colored lines), which we intended to suggest being compared. To avoid confusion, we reformulated in the manuscript:**

**Ln 412: "..., the resulting single-column profiles are in general in good agreement with the full-domain QVPs (compare both for the B-PRO$_{def}$ setup in Figs. 11 and 12), ..."**

- L430: Could the authors speculate as to why the ARlow + σlow could lead to a reduction in ρHV? I would think that a low aspect ratio and low canting angle would lead to more uniform behaviour and hence a larger ρHV.

**We thank the reviewer for this particularly interesting question that made us revisit literature and think again and more deeply about our forward operator assumptions and results.**

**Low aspect ratio (note: following Ryzhkov et al. (2011) we define aspect ratio of the oblates as ratio of the semi-minor to semi-major axes), i.e. a higher degree of nonsphericity, generally leads to a decorrelation of the horizontally and vertically polarized signal returns, i.e. to a reduction in $\rho_{HV}$ (e.g. Kumjian, 2013; Melnikov, 2011). This behaviour is observed in our simulations.**

**Higher degree of orientation, i.e. lower widths of the canting angle distribution, is expected to cause a more uniform effective appearance of the scatterers, i.e. higher correlation coefficients. For the cloud ice sensitivity cases of the CTRL run, our results indeed deviate from that expectation, while it is fulfilled in EXP3.**

**Considering that assumption of homogeneous effective-density particles with a deterministic (or even constant) size-shape relation, and oblate spheroids in particular (see e.g. Zrnic, 1994), creates artificial uniformity in the bulk, hence is a notoriously inept assumption, we prefer(red) to not discuss $\rho_{HV}$ further in the manuscript.**

**For the same reason, we did not analyze the origin of that unexpected behaviour further. However, one possible explanation might be the coexistence of several hydrometeor classes, specifically of snow beside the cloud ice. Both classes are individually rather uniform, but create diversity in the bulk when appearing together. Specifically the $AR_{low}$ setup for ice introduces the biggest differences compared to snow, i.e. creates higher diversity in the bulk compared to the other ice sensitivity setups. Also, the canting angle distributions differ the most between the default snow and the $\sigma_{low}$ setups, which might - other than initially expected - create a higher degree of diversity in the bulk.**

**As pointed out before, we did not test this hypothesis. However, it is in line with the absence of this behaviour in the EXP3 case, where no (or very little) snow or other hydrometeor class than cloud ice is present at these heights.**

References:

Morrison, H., J.A Milbrandt, 2015: Parameterization of Cloud Microphysics Based on the Prediction of Bulk Ice Particle Properties. Part I: Scheme Description and Idealized Tests, Journal of the Atmospheric Sciences, 72, 287-311.

**Included.**

**References:**

Kumjian, M. R.: Principles and applications of dual-polarization weather radar. Part I: Description of the polarimetric radar variables, Journal of Operational Meteorology, 1, 226–242, https://doi.org/10.15191/nwajom.2013.0119, 2013.

Melnikov, V.: Polarimetric properties of ice cloud particles, in: 35th Conference on Radar Meteorology, p. 75, American Meteorological Society, ams.confex.com/ams/35Radar/webprogram/Paper191041.html, 2011.

Ryzhkov, A., Pinsky, M., Pokrovsky, A., and Khain, A.: Polarimetric Radar Observation Operator for a Cloud Model with Spectral Microphysics, Journal of Applied Meteorology and Climatology, 50, 873–894, https://doi.org/10.1175/2010JAMC2363.1, 2011.

Zrnic, D. S., Balakrishnan, N., Ryzhkov, A. V., and Durden, S. L.: Use of copolar correlation coefficient for probing precipitation at nearly vertical incidence, IEEE Transactions on Geoscience and Remote Sensing, 32, 740–748, https://doi.org/10.1109/36.298003, 1994.

---

## Author Comment (AC2)

General: I think this is a nice study demonstrating the uncertainties not only with model microphysics parameterizations, but also the challenges with polarimetric radar forward operators. I have a few minor comments.

**We are very thankful for the reviewer's comments. Below, we address the reviewer's specific comments (in bold blue).**

Abstract: A reader approaching this for the first time does not know what Dice or Tgr are. Secondly, it is unclear what a 'low bias' is in the sense of polarimetric moments.

**The text in the abstract has been modified for clarity:**

**Ln 5: "Modifying the critical diameter of particles for ice-to-snow conversion by aggregation (Dice) and the threshold temperature responsible for graupel production by riming (Tgr), was found to improve the synthetic polarimetric moments and simulated hydrometeor population, while keeping the difference in surface precipitation statistically insignificant at model resolvable grid scales. However, the model still exhibited a low bias (lower magnitude than observation)..."**

Ln28 – remove extra ')'.

**Corrected.**

Ln 34: "..total mass peak" what is meant here? The model was not able to capture the correct height of the ice mass or the ice mass was too large?

**Here, the "mass peak" refers to the peak of ice mass size distribution. The sentence has been rephrased in the revised manuscript for clarity.**

**Ln 37: "…model underpredicted total ice number concentrations and overpredicted the peak of mass size distribution…"**

Ln 75 – I am confused about what 99.9 m MSL is indicating here. Is that the actual height of the BoXPol radar?

**Yes, it is the height of the BoXPol radar with reference to the mean sea level (MSL). The sentence has been rephrased for clarity: Ln XXX: "…, University of Bonn at 99.9 m above MSL".**

Ln 125: "it μ depends on" -> "it's μ depends on"

**Corrected.**

Ln 130: What is 'n'?

**Here, "n" is the specific hydrometeor number, which is the ratio of hydrometeor number density to total density (air, vapor, hydrometeors). It is defined as such in the earlier paragraph.**

Ln 149: I'm curious about the selection of Dice for the sensitivity studies. The range of values from 5.0 to 400.0 seem like rather large perturbations, and 400.0 seems excessive compared to the original 50.0 . I suppose such a large value essentially limits the production of snow from ice? How were these values selected for the sensitivity tests?

**We conducted a sensitivity study using multiple values of Dice (e.g., 5, 50, 150, 400, 800 $\mu m$). And, for the upper extent from the default value, Dice = 400 $\mu m$ showed considerable improvement in the synthetic polarimetric signatures at upper levels, and is used in this study. With Dice=800 $\mu m$, it starts to limit the production of snow from ice.**

**The following paragraph has been added in the revised manuscript to clarify the setup of model sensitivity study:**

**Ln 335: "For the cloud ice aggregation threshold, we conducted a sensitivity study using multiple values of Dice (e.g., 5, 50, 150, 400, 800 $\mu m$), the default value being 50 $\mu m$. For brevity, we only report on the results from one lower and one upper value as well as the default value. From these experiments, Dice = 400 $\mu m$ showed the best improvement in the synthetic polarimetric signatures and is used as the upper Dice value in this study. Similarly, we varied Tgr from the default 0°C by reducing it by 5 and 3 °C respectively, to check the sensitivity of graupel production near the melting layer. The four experiments together constitute different combinations of aggregation (ice/snow partitioning) and riming (graupel production and rain gradient below melting layer)."**

Ln 178: Put () around 2 to be consistent.

**Added.**

Ln 224: What temperatures (are they derived from a sounding, model, etc.) are used for the HCA-Dolan and HCA-Zrnic?

**For HCA-Dolan and HCA-Zrnic the COSMO model temperatures were used. These were used for all calculations on the radar side to remain consistent.**

Ln 227: "Problem description" seems a little ominous. Perhaps "Case description?"

**We have changed it to "Case Description" in the revised manuscript.**

Ln 230: Spell out 'approx.' to approximation.

**Corrected.**

Ln 308: I am confused here about the inter-changed language of mixing ratio and percentages. Are you able to derive an actual mixing ratio (g/kg) for the different hydrometeor types with HCA-Pejcic? Or is this more related to the probability of a given hydrometeor type within a volume?

**It is not possible for us to calculate a mixing ratio in g/kg. What we can derive is the hydrometeor partitioning (PR) from the radar measurements. This is a relative mass contribution of specific hydrometeor types in %. The distance between a multidimensional measurement, in our case a vector consisting of horizontal reflectivity, differential reflectivity, specific differential phase, cross correlation coefficient and a temperature indicator, and a hydrometeor specific centroid derived from a clustering procedure is interpreted as PR. More details on the method can be found in Besic et al. 2018 (Unraveling hydrometeor mixtures in polarimetric radar measurements, https://doi.org/10.5194/amt-11-4847-2018) and on our refinement in Pejcic et al. 2021 (Polarimetric radar-based methods for evaluation of hydrometeor mixtures in numerical weather prediction models, https://doi.org/10.23919/IRS51887.2021.9466201).**

**For clarity, the "mixing ratio" in Section 3.2 has been rephrased to hydrometeor percentage.**

Ln 357-360: Unless I'm missing something in the figures, the mean sizes are not shown? That is fine, but perhaps add (not shown) so the reader is not feeling like they are missing something from the figures.

**Yes, for brevity, the figures of mean sizes are not shown. The sentence has been modified in the revised manuscript for clarity:**

**Ln 370:** " …the mean size of graupel around the vicinity of the melting layer is around 1.5 - 2 mm for CTRL and EXP1 (not shown here)."

Ln 424: This reads a little strange with "Since". Perhaps it would make more sense to start with "Even though…"

**Reformulated.**

Ln 430: It is hard to cross-reference these temperature on Fig. 11. Perhaps add a temperature scale on the right side as in Fig. 9?

**To clarify, we added the respective information in the manuscript text:**

**Ln 444: "The snow-dominated lower levels (-3 to -13°C, approximately corresponding here to heights of 2.0 to 4.0 km) are characterized by a strong underestimation of $Z_{DR}$.."**

Ln 465: If I'm not mistaken, Dice was increased in EXP1,3 not EXP2,3?

**Yes, Dice was increased in EXP 1,3. Fixed.**

Ln 489: EMA has not been defined.

**We spell out the acronym here now:**

**Ln 503: "Concerning the effective medium approximation (EMA) for the ice-air mixture material of dry snowflakes…"**

Figure 3: Perhaps state in the figure caption that the top row is the CTRL and the bottom is EXP3 for clarity.

**Added.**

Figure 6: I understand the logic of keeping the color scales consistent for each HCA, but it is a bit confusing that the different colors do not align with different species across the three HCAs. For example, it is great that RN and 'snow/Aggregates' are the same colors across all three, but melting snow is purple in a) and cyan in b) and c); light blue is ice crystals in a) and light rain/drizzle in b) and c), and perhaps most confusingly, graupel is orange in a) but ice crystals and vertical ice in c). I also find the purple for WS difficult to distinguish from the blue RN and black HA in a). Finally, in the caption for panel c) vertical ice is listed as VI but in the colorbar it is VC.

**We have now adjusted the color bars. Wet Snow is now "cyan" in all three plots, Ice is now also grey in a) as in b) and c). Graupel in a) and Low Density Sleet (LG) in b) are now in the colour 'Fuchsia' like Graupel/Hail (GH) in c). The caption for panel c) has been changed from VI to VC, matching with the colorbar.**

[Figure]

Table 1: are [m s-3] the correct units for vT?

**Corrected to [m s$^{-1}$].**

Table 2: The values for EMA are meaningless to me unless I dig up Blahak (2016). Is there a simple way to describe what these mean?

**In the table 2 caption, we replaced the puristic reference to Blahak (2016) and added a short explanation of the three different EMA applied in the setup used in the study:**

**"The detailed meaning of the EMA and a complete overview over all options is given in Blahak (2016). All three settings applied here make use of the Maxwell-Garnett mixing rule (Maxwell Garnett, 1905). 'mas': ice-air mixture with air as matrix and spheroidal inclusions of ice. 'mis': similar, but with ice as matrix and air as inclusions. 'mawsms': a three-component (ice-water-air) mixture constructed as a two-fold two-component mixture, where spheroidal air inclusions are suspended in an ice-water matrix, the latter with spheroidal ice inclusions in a water matrix."**